# LIPSCHITZ LIFELONG REINFORCEMENT LEARNING

## ABSTRACT

We consider the problem of knowledge transfer when an agent is facing a series of Reinforcement Learning (RL) tasks. We introduce a novel metric between Markov Decision Processes and establish that close MDPs have close optimal value functions, that is that optimal value functions are Lipschitz continuous with respect to tasks. These theoretical results lead us to a value transfer method for Lifelong RL, which we use to build a PAC-MDP algorithm with improved convergence rate. We illustrate the benefits of the method in Lifelong RL experiments.

## 1 INTRODUCTION

Lifelong Reinforcement Learning (RL) is an online problem where an agent faces a series of RL tasks, drawn sequentially. Transferring the knowledge of prior experience while solving new tasks is a key question in that setting (Lazaric, 2012; Taylor and Stone, 2009). We elaborate on the intuitive idea that *similar* tasks should allow a large amount of transfer. An agent able to compute online a similarity measure between source tasks and the current target task should be able to perform transfer. By measuring the amount of inter-task similarity, we design a novel method for value transfer, practically deployable in the online Lifelong RL setting. Specifically, we introduce a metric between MDPs and prove that the optimal Q-value function is Lipschitz continuous with respect to MDPs. This property allows to compute a provable upper-bound on the optimal value function of a target task, given the learned optimal value function of a source task. Knowing this upper bound allows to accelerate the convergence of an RMax algorithm (Brafman and Tennenholtz, 2002). This transfer method is non-negative (it cannot cause performance degradation) as the computed upper bound does not underestimate the optimal Q-value function.

Our contributions are as follows. First, we study theoretically the Lipschitz continuity of the optimal value function in the task space (Section 3). Then, we use this continuity property to propose a value-transfer method based on a local distance between MDPs (Section 4). Full knowledge of both MDPs is not required and the transfer is non-negative, which makes the method both practical and safe. In Section 4.2, we build a PAC-MDP algorithm called *Lipschitz RMax*, applying this transfer method online in the Lifelong RL setting. We provide sample and computational complexity bounds and showcase the algorithm in Lifelong RL experiments (Section 5).

## 2 BACKGROUND AND RELATED WORK

Reinforcement Learning (RL) (Sutton and Barto, 1998) is a framework for sequential decision making. The problem is typically modeled as a Markov Decision Process (MDP) (Puterman, 2014) consisting in a 4-tuple $\langle \mathcal{S}, \mathcal{A}, R, T \rangle$ where $\mathcal{S}$ is a state space, $\mathcal{A}$ an action space, $R_s^a$ is the expected reward of taking action $a$ in state $s$ and $T_{ss'}^a$ is the transition probability of reaching state $s'$ when taking action $a$ in state $s$. Without loss of generality, we assume $R_s^a \in [0, 1]$. Given a discount factor $\gamma \in [0, 1)$, the expected cumulative return $\sum_t \gamma^t R_{s_t}^{a_t}$ obtained along a trajectory starting with state $s$ and action $a$ is noted $Q(s, a)$ and called the Q-function. The optimal Q-function $Q^*$ is the highest attainable expected return from $s, a$ and $V^*(s) = \max_{a \in \mathcal{A}} Q^*(s, a)$ is the optimal value function in $s$.

Lifelong RL (Silver et al., 2013; Brunskill and Li, 2014) is the problem of experiencing online a series of MDPs drawn from an unknown distribution. Each time an MDP is sampled, a classical RL problem takes place where the agent is able to interact with the environment to maximize its expected return. In this setting, it is reasonable to think that knowledge gained on previous MDPs could be re-used to improve the performance in new MDPs. In this paper, we provide a novel method for such

transfer by characterizing the way the optimal Q-function can evolve across tasks. We restrict the scope of the study to the case where sampled MDPs share the same state-action space $\mathcal{S} \times \mathcal{A}$. For brevity, we will refer indifferently to MDPs, models or tasks, and write them $M = \langle R, T \rangle$.

Using a metric between MDPs has the appealing characteristic of quantifying the amount of similarity between tasks, which intuitively should be linked to the amount of transfer achievable. Song et al. (2016) define a metric based on the bi-simulation metric introduced by Ferns et al. (2004) and the Wasserstein metric (Villani, 2008). Value transfer is performed between states with low bi-simulation distances. However, this metric requires knowing both MDPs completely and is thus unusable in the Lifelong RL setting where we expect to perform transfer before having learned the current MDP. Further, the transfer technique they propose does allow negative transfer (see Appendix, Section A). Carroll and Seppi (2005) also define a value-transfer method based on a measure of similarity between tasks. However, this measure is not computable online and thus not applicable to the Lifelong RL setting. Mahmud et al. (2013) and Brunskill and Li (2013) propose MDP clustering methods respectively using a metric quantifying the regret of running the optimal policy of one MDP in the other MDP and the $\mathcal{L}_1$ norm between the MDP models. An advantage of clustering is to prune the set of possible source tasks. They use their approach for policy transfer, which differs from the value-transfer method proposed in this paper. Ammar et al. (2014) learn the model of a source MDP and view the prediction error on a target MDP as a dissimilarity measure in the task space. Their method makes use of samples from both tasks and is not readily applicable to the online setting considered in this paper. Lazaric et al. (2008) provide a practical method for sample transfer, computing a similarity metric reflecting the probability of the models to be identical. Their approach is applicable in a batch RL setting as opposed to our online setting. The approach developed by Sorg and Singh (2009) is very similar to ours in the sense that they prove bounds on the optimal Q-function for new tasks, assuming that both MDPs are known and that a soft homomorphism exists between the state spaces. Brunskill and Li (2013) also provide a method that can be used for Q-function bounding in multi-task RL. Abel et al. (2018) present the MaxQInit algorithm, providing transferred bounds on the Q-function with high probability while preserving PAC-MDP guarantees. Given a set of solved tasks, they derive the probability that the maximum over the Q-values of previous MDPs is an upper bound on the current task's optimal Q-function. This results in a method for non-negative transfer with high probability once enough tasks have been sampled.

## 3 LIPSCHITZ CONTINUITY OF Q-FUNCTIONS

The intuition we build on is that similar MDPs should have similar optimal Q-functions. Formally, this insight can be translated into a continuity property of the optimal Q-functions over the MDP space $\mathcal{M}$. The remainder of this section mathematically formalizes this intuition that will be used in the next Section to derive a practical method for value transfer. To that end, we introduce a local pseudo-metric characterizing the distance between the models of two MDPs at a particular state-action pair. A reminder and a detailed discussion on the metrics (and related objects) used herein can be found in the Appendix, Section B.

**Definition 1.** *Given two tasks $M = \langle R, T \rangle$ and $\bar{M} = \langle \bar{R}, \bar{T} \rangle$, and a function $f : \mathcal{S} \to \mathbb{R}^+$, we define the* pseudo-metric between models *at $(s, a) \in \mathcal{S} \times \mathcal{A}$ w.r.t. $f$ as:*

$$D_f^{M\bar{M}}(s, a) \triangleq |R_s^a - \bar{R}_s^a| + \sum_{s' \in \mathcal{S}} f(s')|T_{ss'}^a - \bar{T}_{ss'}^a|. \tag{1}$$

This pseudo-metric is relative to a positive function $f$. We implicitly cast this definition in the context of discrete state spaces. The extension to continuous spaces is straightforward but beyond the scope of this paper. Let $Q_M^*$ denote the optimal Q-function of MDP $M \in \mathcal{M}$.

**Proposition 1** (Local pseudo-Lipschitz continuity). *For two MDPs $M, \bar{M}$, for all $(s, a) \in \mathcal{S} \times \mathcal{A}$,*

$$|Q_M^*(s, a) - Q_{\bar{M}}^*(s, a)| \le \Delta^{M\bar{M}}(s, a), \tag{2}$$

*with the* MDPs local pseudo-metric $\Delta^{M\bar{M}}(s, a) \triangleq \min\left\{d_M^{\bar{M}}(s, a), d_{\bar{M}}^M(s, a)\right\}$, *and the* local MDP dissimilarity $d_M^{\bar{M}} : \mathcal{S} \times \mathcal{A} \to \mathbb{R}$ *is the unique solution to the following fixed-point equation for d:*

$$d(s, a) = D_{\gamma V_{\bar{M}}^*}^{M\bar{M}}(s, a) + \gamma \sum_{s' \in \mathcal{S}} T_{ss'}^a \max_{a' \in \mathcal{A}} d(s', a'). \tag{3}$$

All the proofs of the paper can be found in the Appendix. This result establishes that the distance between the optimal Q-functions of two MDPs at $(s, a) \in \mathcal{S} \times \mathcal{A}$ is controlled by a local dissimilarity between the MDPs. The latter follows a fixed-point equation (Equation 3), which can be solved by Dynamic Programming (DP) (Bellman, 1957). Note that, although the local MDP dissimilarity $d_M^{\bar{M}}$ is asymmetric, $\Delta^{M\bar{M}}(s, a)$ *is* a pseudo-metric, hence the name *pseudo-Lipschitz continuity*. Similar results for the value function of a fixed policy and the optimal value function $V_M^*$ can easily be derived, as well as a global pseudo-Lipschitz continuity property (Appendix, Sections D and E). Thus, overall, the optimal Q-functions of two close MDPs (in the sense of Equation 1) are themselves close to each other. A direct consequence in Lifelong RL is that previously solved tasks can help bound the value function of a new task. Even a partially learned Q-function can be used for that purpose if error bounds are known or if it provably overestimates the true $Q^*$. The next section exploits this property to build a PAC-MDP algorithm that performs provably non-negative transfer between successive tasks and accelerates learning.

## 4 TRANSFER USING THE LIPSCHITZ CONTINUITY

A purpose of value transfer, when interacting online with a new MDP, is to initialize the value function and drive the exploration to accelerate learning. We aim to exploit value transfer in a method guaranteeing three conditions: C1. the resulting algorithm is PAC-MDP (Strehl et al., 2009); C2. the transfer accelerates learning; C3. the transfer is non-negative. From Proposition 1, one can naturally define a local upper bound on the optimal Q-function of an MDP given the optimal Q-function of another MDP.

**Definition 2.** *Given two tasks $M$ and $\bar{M}$, for all $(s, a) \in \mathcal{S} \times \mathcal{A}$, the* Lipschitz upper bound *on $Q_M^*$ induced by $Q_{\bar{M}}^*$ is defined as $U_{\bar{M}}(s, a) \geq Q_M^*(s, a)$ with:*

$$U_{\bar{M}}(s, a) \triangleq Q_{\bar{M}}^*(s, a) + \Delta^{M\bar{M}}(s, a). \tag{4}$$

The *optimism in the face of uncertainty* principle leads to consider that the long-term expected return from any state is the $\frac{1}{1-\gamma}$ maximum return, unless proven otherwise. The RMax algorithm (Brafman and Tennenholtz, 2002) in particular explores an MDP so as to shrink this upper bound. RMax is a model-based, online RL algorithm with PAC-MDP guarantees (Strehl et al., 2009) which means that convergence to near-optimal policy is guaranteed in a polynomial number of steps. It relies on an optimistic model initialization that yields an optimistic upper bound $U$ on the optimal Q-function, then acts greedily w.r.t. $U$. By default, it takes the maximum value $U(s, a) = \frac{1}{1-\gamma}$ but any tighter upper bound is admissible. Thus, shrinking $U$ with Equation 4 is expected to improve the learning speed for new tasks in Lifelong RL.

In RMax, during the resolution of a task $M$, $\mathcal{S} \times \mathcal{A}$ is split into a subset of known state-action pairs $K$ and its complement $K^c$ of unknown pairs. A state-action pair is known if the number of collected reward and transition samples allows estimating an $\epsilon$-accurate model in $\mathcal{L}_1$-norm with probability higher than $1 - \delta$. We refer to $\epsilon$ and $\delta$ as the *RMax precision parameters*. This translates into a threshold $n_{known}$ on the number of visits $n(s, a)$ to a pair $s, a$ that are necessary to reach this precision. Given the experience of a set of $m$ MDPs $\bar{\mathcal{M}} = \{\bar{M}_1, \ldots, \bar{M}_m\}$, we define the total bound as the minimum over all the Lipschitz bounds induced by each previous MDP.

**Proposition 2.** *Given a partially known task $M = \langle R, T \rangle$, the set of known state-action pairs $K$, and the set of Lipschitz bounds on $Q_M^*$ induced by previous tasks $\{U_{\bar{M}_1}, \ldots, U_{\bar{M}_m}\}$, the function $Q$ defined below is an upper bound on $Q_M^*$ for all $s, a \in \mathcal{S} \times \mathcal{A}$.*

$$Q(s, a) \triangleq \begin{cases} R_s^a + \gamma \sum_{s' \in \mathcal{S}} T_{ss'}^a \max_{a'} Q(s', a') & \text{if } (s, a) \in K, \\ U(s, a) & \text{otherwise,} \end{cases} \tag{5}$$

*with $U(s, a) = \min\left\{\frac{1}{1-\gamma}, U_{\bar{M}_1}(s, a), \ldots, U_{\bar{M}_m}(s, a)\right\}$.*

Traditionally in RMax, Equation 5 is solved to a precision $\epsilon_Q$ via Value Iteration. This yields a function $Q$ that is a valid heuristic (provable upper bound on $Q_M^*$) for the exploration of MDP $M$.

## 4.1 A TRACTABLE UPPER BOUND ON $Q_M^*$

The key issue addressed in this Section is how to actually compute $U(s, a)$. Consider two tasks $M$ and $\bar{M}$, on which vanilla RMax has been applied, yielding the respective sets of known state-action pairs $K$ and $\bar{K}$, along with the learned models $\hat{M} = \langle \hat{T}, \hat{R} \rangle$ and $\hat{\bar{M}} = \langle \hat{\bar{T}}, \hat{\bar{R}} \rangle$, and the upper bounds $Q$ and $\bar{Q}$ respectively on $Q_M^*$ and $Q_{\bar{M}}^*$. Equation 5 allows the transfer of knowledge from $\bar{M}$ to $M$ if $U_{\bar{M}}(s, a)$ can be computed. Unfortunately, the true optimal value functions, transition and reward models, necessary to compute $U_{\bar{M}}$, are unknown. Thus, we propose to compute a looser upper bound based on the learned models and value functions. First, we provide an upper bound $\hat{D}^{M\bar{M}}$ on the pseudo metric between models $M$ and $\bar{M}$.

**Proposition 3.** *Given two tasks $M$ and $\bar{M}$, $K$ and $\bar{K}$ the respective sets of state-action pairs where their models are known with accuracy $\epsilon$ in $\mathcal{L}_1$-norm with probability at least $1 - \delta$,*

$$\boldsymbol{Pr}\left(\hat{D}^{M\bar{M}}(s, a) \geq D_{\gamma V_{\bar{M}}^*}^{M\bar{M}}(s, a)\right) \geq 1 - \delta$$

*with the following definition of the* upper bound on the pseudo-metric between models $\hat{D}^{M\bar{M}}$:

$$\hat{D}^{M\bar{M}}(s, a) \triangleq \begin{cases} D_{\gamma\bar{V}}^{\hat{M}\hat{\bar{M}}}(s, a) + 2B & \text{if } (s, a) \in K \cap \bar{K} \\ \max_{\bar{\mu} \in \mathcal{M}} D_{\gamma\bar{V}}^{\hat{M}\bar{\mu}}(s, a) + B & \text{if } (s, a) \in K \cap \bar{K}^c \\ \max_{\mu \in \mathcal{M}} D_{\gamma\bar{V}}^{\mu\hat{\bar{M}}}(s, a) + B & \text{if } (s, a) \in K^c \cap \bar{K} \\ \max_{\mu, \bar{\mu} \in \mathcal{M}^2} D_{\gamma\bar{V}}^{\mu\bar{\mu}}(s, a) & \text{if } (s, a) \in K^c \cap \bar{K}^c \end{cases} \tag{6}$$

*where $B = \epsilon\left(1 + \gamma \max_{s'} \bar{V}(s')\right)$.*

This upper bound $\hat{D}^{M\bar{M}}$ on the distance between MDPs can be calculated analytically (see Appendix, Section H). The magnitude of the $B$ term is controlled by $\epsilon$. In the case where no information is available on the maximum value of $\bar{V}$, $B = \frac{\epsilon}{1-\gamma}$. $\epsilon$ measures the accuracy with which the tasks are known: the smaller $\epsilon$, the tighter the $B$ bound. Note that $\bar{V}$ is used as an upper bound on the true $V_{\bar{M}}^*$. In many cases, $\max_{s'} V_{\bar{M}}^*(s') \ll \frac{1}{1-\gamma}$; *e.g.* for stochastic shortest path problems, which feature rewards only upon reaching terminal states, $\max_{s'} V_{\bar{M}}^*(s') = 1$ and thus $B = (1 + \gamma)\epsilon$ is a tighter bound for transfer. Using $\hat{D}^{M\bar{M}}$ and Equation 3, one can derive an upper bound $\hat{d}_M^{\bar{M}}$ on $d_M^{\bar{M}}$, detailed in Proposition 4.

**Proposition 4.** *Given two tasks $M$ and $\bar{M}$, $K$ the set of state-action pairs for which $\langle R, T \rangle$ is known with accuracy $\epsilon$ in $\mathcal{L}_1$-norm with probability at least $1 - \delta$. If $\gamma(1 + \epsilon) < 1$, the solution $\hat{d}_M^{\bar{M}}$ of the following fixed-point equation on $\hat{d}$ is an upper bound on $d_M^{\bar{M}}$ with probability at least $1 - \delta$:*

$$\hat{d}(s, a) = \hat{D}^{M\bar{M}}(s, a) + \begin{cases} \gamma\left(\sum_{s' \in \mathcal{S}} \hat{T}_{ss'}^a \max_{a' \in \mathcal{A}} \hat{d}(s', a') + \epsilon \max_{s', a' \in \mathcal{S} \times \mathcal{A}} \hat{d}(s', a')\right) & \text{if } s, a \in K, \\ \gamma \max_{s', a' \in \mathcal{S} \times \mathcal{A}} \hat{d}(s', a') & \text{otherwise.} \end{cases} \tag{7}$$

Similarly as in Proposition 3, the condition $\gamma(1 + \epsilon) < 1$ illustrates the fact that for a large return horizon (large $\gamma$), a high accuracy (small $\epsilon$) is needed for the bound to be computable. Finally, a tractable upper bound on $Q_M^*$ given $\bar{M}$ with high probability is given by

$$\hat{U}_{\bar{M}}(s, a) = \bar{Q}(s, a) + \min\left\{\hat{d}_M^{\bar{M}}(s, a), \hat{d}_{\bar{M}}^M(s, a)\right\}. \tag{8}$$

And the associated upper bound on $U(s, a)$ (Equation 5) given previous tasks $\bar{\mathcal{M}} = \{\bar{M}_i\}_{i=1}^m$ is

$$\hat{U}(s, a) = \min\left\{\frac{1}{1-\gamma}, \hat{U}_{\bar{M}_1}(s, a), \dots, \hat{U}_{\bar{M}_m}(s, a)\right\} \tag{9}$$

This upper bound can be used to transfer knowledge from a partially solved task to a target task. If $\hat{U}(s, a) \leq \frac{1}{1-\gamma}$ for some $(s, a)$ pairs, then the convergence rate can be improved. As complete knowledge of both tasks is not needed, it can be applied online in a Lifelong RL setting. In the next section, we explicit an algorithm that leverages this value transfer method.

---

**Algorithm 1:** Lipschitz RMax algorithm

---

Initialize $\hat{\mathcal{M}} = \emptyset$.
**for** *each newly sampled MDP $M$* **do**
    Initialize $Q(s, a) = \frac{1}{1-\gamma}, \forall s, a$, and $K = \emptyset$
    Initialize $\hat{T}$ and $\hat{R}$ (RMax initialization)
    $Q \leftarrow \text{UpdateQ}(\hat{\mathcal{M}}, \hat{T}, \hat{R})$
    **for** $t \in [1, \text{max number of steps}]$ **do**
        $s = \text{current state}, a = \arg\max_{a'} Q(s, a')$
        Observe reward $r$ and next state $s'$
        $n(s, a) \leftarrow n(s, a) + 1$
        **if** $n(s, a) < n_{known}$ **then**
            Store $(s, a, r, s')$
        **if** $n(s, a) = n_{known}$ **then**
            Update $K$, $\hat{T}^a_{ss'}$ and $\hat{R}^a_s$
            $Q \leftarrow \text{UpdateQ}(\hat{\mathcal{M}}, \hat{T}, \hat{R})$
    Save $\hat{M} = \left(\hat{T}, \hat{R}, K, Q\right)$ in $\hat{\mathcal{M}}$

Function UpdateQ($\hat{\mathcal{M}}, \hat{T}, \hat{R}$):
**for** $\bar{M} \in \bar{\mathcal{M}}$ **do**
    Compute $\hat{D}^{M\bar{M}}$ and $\hat{D}^{\bar{M}M}$ (Eq. 6)
    Compute $\hat{d}^{\bar{M}}_M$ and $\hat{d}^M_{\bar{M}}$ (DP on Eq. 7)
    Compute $\hat{U}_{\bar{M}}$ (Eq. 8)
Compute $\hat{U}$ (Eq. 9)
Compute and return $Q$ (DP on Eq. 5 using $\hat{U}$)

---

## 4.2 LIPSCHITZ RMAX

In Lifelong RL, MDPs are encountered sequentially. Applying RMax to task $M$ yields the set of known state-action pairs $K$, the learned models $\hat{T}$ and $\hat{R}$, and the upper bound $Q$ on $Q^*_M$. Saving this information when the task changes allows to compute the upper bound of Equation 9 for the new task, and to use it to shrink the optimistic heuristic of RMax. This effectively transfers value functions between tasks based on task similarity. As the new task is explored online, the task similarity is progressively assessed with better confidence, refining the values of $\hat{D}^{M\bar{M}}$, $\hat{d}^{\bar{M}}_M$ and eventually $\hat{U}$, allowing for more efficient transfer where the task similarity is appraised. The resulting algorithm, Lipschitz RMax (LRMax), is presented in Algorithm 1. To avoid ambiguities with $\bar{\mathcal{M}}$, we use $\hat{\mathcal{M}}$ to store learned features ($\hat{T}$, $\hat{R}$, $K$, $Q$) about previous MDPs. In a nutshell, the behavior of LRMax on a given task $M$ is precisely that of RMax, but with a tighter admissible heuristic $\hat{U}$ that becomes better as the new task is explored (while this heuristic remains constant in vanilla RMax). LRMax is PAC-MDP (Condition C1) as stated in Propositions 5 and 6 below. With $S = |\mathcal{S}|$ and $A = |\mathcal{A}|$, the sample complexity of vanilla RMax is $\tilde{\mathcal{O}}(S^2 A/(\epsilon^3 (1-\gamma)^3))$, which is improved by LRMax in Proposition 5 and meets Condition C2. Finally $\hat{U}$ is a proved upper bound with high probability on $Q^*_M$, which avoids negative transfer and meets Condition C3.

**Proposition 5** (Sample complexity (Strehl et al., 2009))**.** *With probability $1 - \delta$, the greedy policy w.r.t. $Q$ computed by LRMax achieves an $\epsilon$-optimal return in MDP $M$ after*

$$\tilde{\mathcal{O}}\left(\frac{S|\{s, a \in \mathcal{S} \times \mathcal{A} \mid \hat{U}(s, a) \geq V^*_M(s) - \epsilon\}|}{\epsilon^3 (1-\gamma)^3}\right)$$

*samples (when logarithmic factors are ignored), with $\hat{U}$ defined in Equation 9 a non-static, decreasing quantity, upper bounded by $\frac{1}{1-\gamma}$.*

Consequently from Proposition 5, the sample complexity of LRMax is no worse than that of RMax.

**Proposition 6** (Computational complexity)**.** *The total computational complexity of Lipschitz RMax is*

$$\tilde{\mathcal{O}}\left(\tau + \frac{S^2 A^2 (S + \log(A))(2N + 1)}{(1-\gamma)} \log\frac{1}{\epsilon_Q (1-\gamma)}\right)$$

*with $\tau$ the number of interaction steps, $\epsilon_Q$ the precision of value iteration and $N$ the number of tasks.*

### 4.3 REFINING LRMAX BOUNDS WITH MAXIMUM MODEL DISTANCE

LRMax relies on upper bounds on the local distances between tasks (Equation 7). The quality of the Lipschitz bound on $Q_M^*$ greatly depends on the quality of those estimates and can be improved accordingly. We discuss two methods to provide finer estimates.

First, from the definition of $D_{\gamma V_{\bar{M}}^*}^{M\bar{M}}(s,a)$, it is easy to show that model pseudo-distances are always upper bounded by $\frac{1+\gamma}{1-\gamma}$. However, in practice, the tasks experienced in Lifelong RL might not cover the full span of possible MDPs and may be systematically closer to each other than $\frac{1+\gamma}{1-\gamma}$. For instance, the distance between two games in the Arcade Learning Environment (ALE) (Bellemare et al., 2013), is smaller than the maximum distance between any two MDPs defined on the common state-action space of the ALE (extended discussion in Appendix, Section L). Let $D_{\max}(s,a) \triangleq \max_{M,\bar{M}\in\mathcal{M}^2}\{D_{\gamma V_{\bar{M}}^*}^{M\bar{M}}(s,a)\}$ be the *maximum model distance* at a particular $s,a$ pair. *Prior knowledge* might indicate a smaller upper bound for $D_{\max}(s,a)$ than $\frac{1+\gamma}{1-\gamma}$. We will note such an upper bound $D_{\max}$. Solving Equation 7 boils down to accumulating $\hat{D}^{M\bar{M}}(s,a)$ values in $\hat{d}(s,a)$. Reducing a $\hat{D}^{M\bar{M}}(s,a)$ estimate in a single $(s,a)$ pair actually reduces $\hat{d}(s,a)$ in *all* $(s,a)$ pairs. Thus, replacing $\hat{D}^{M\bar{M}}(s,a)$ in Equation 7 by $\min\{D_{\max}, \hat{D}^{M\bar{M}}(s,a)\}$, provides a smaller upper bound $\hat{d}_M^{\bar{M}}$ on $d_M^{\bar{M}}$, and thus a smaller $\hat{U}$ which allows transfer if it is lesser than $\frac{1}{1-\gamma}$. Consequently, such an upper bound $D_{\max}$ can make a difference between successful and unsuccessful transfer, even if its value is of little importance. Conversely, setting a value for $D_{\max}$ quantifies the distance between MDPs where transfer is efficient.

Furthermore, one can estimate online the value of $D_{\max}(s,a)$, lifting the previous hypothesis of available prior knowledge. One can build an empirical estimate of the maximum model distance at $s,a$: $\hat{D}_{\max}(s,a) \triangleq \max_{M,\bar{M}\in\hat{\mathcal{M}}^2}\{\hat{D}^{M\bar{M}}(s,a)\}$, $\hat{\mathcal{M}}$ being the set of explored tasks. The pitfall being that, with few explored tasks, $\hat{D}_{\max}(s,a)$ could underestimate $D_{\max}(s,a)$. Proposition 7 provides a lower bound on the probability that $\hat{D}_{\max}(s,a)$ does not underestimate $D_{\max}(s,a)$, depending on the number of sampled tasks. In turn this indicates when $\hat{D}_{\max}(s,a)$ upper bounds $D_{\max}(s,a)$ with high probability, which can be combined with Algorithm 1 to improve the performance.

**Proposition 7.** *Consider an algorithm producing $\epsilon$-accurate in $\mathcal{L}_1$-norm model estimates with probability at least $1-\delta$ for a subset of $\mathcal{S}\times\mathcal{A}$ after interacting with an MDP. For all $s,a\in\mathcal{S}\times\mathcal{A}$, after sampling $m$ tasks with $p_{\min} = \min_{M\in\mathcal{M}} \boldsymbol{Pr}(M)$, the following lower bound holds:*

$$\boldsymbol{Pr}\left(\hat{D}_{\max}(s,a) \geq D_{\max}(s,a)\right) \geq 1 - 2(1-p_{\min})^m + (1-2p_{\min})^m.$$

The assumption of a lower bound $p_{\min}$ on the sampling probability of a task implies that $\mathcal{M}$ is finite and is commonly seen as a non-adversarial task sampling strategy (Abel et al., 2018).

## 5 EXPERIMENTS

The experiments reported here[1] illustrate how 1) LRMax allows for early performance increase in Lifelong RL by efficiently transferring knowledge between tasks; 2) the Lipschitz bound of Equation 8 improves the sample complexity compared to RMax by providing a tighter upper bound on $Q^*$. Graphs are displayed with 95% confidence intervals. Information in line with the Machine Learning Reproducibility Check-list (Pineau, 2019) is documented in the Appendix, Section Q.

We evaluate different variants of LRMax in a Lifelong RL experiment. The RMax algorithm will be used as a no-transfer baseline. LRMax($x$) denotes Algorithm 1 with prior $D_{\max} = x$. MaxQInit denotes the MAXQINIT algorithm from Abel et al. (2018), consisting in a state-of-the art PAC-MDP algorithm achieving transfer with PAC guarantees. Both LRMax and MaxQInit algorithms achieve value transfer by providing a tighter upper-bound on $Q^*$ than $\frac{1}{1-\gamma}$. Computing both upper-bounds and taking the minimum results in combining the two approaches. We include such a combination in

---

[1]Link to open-source code omitted for anonymity.

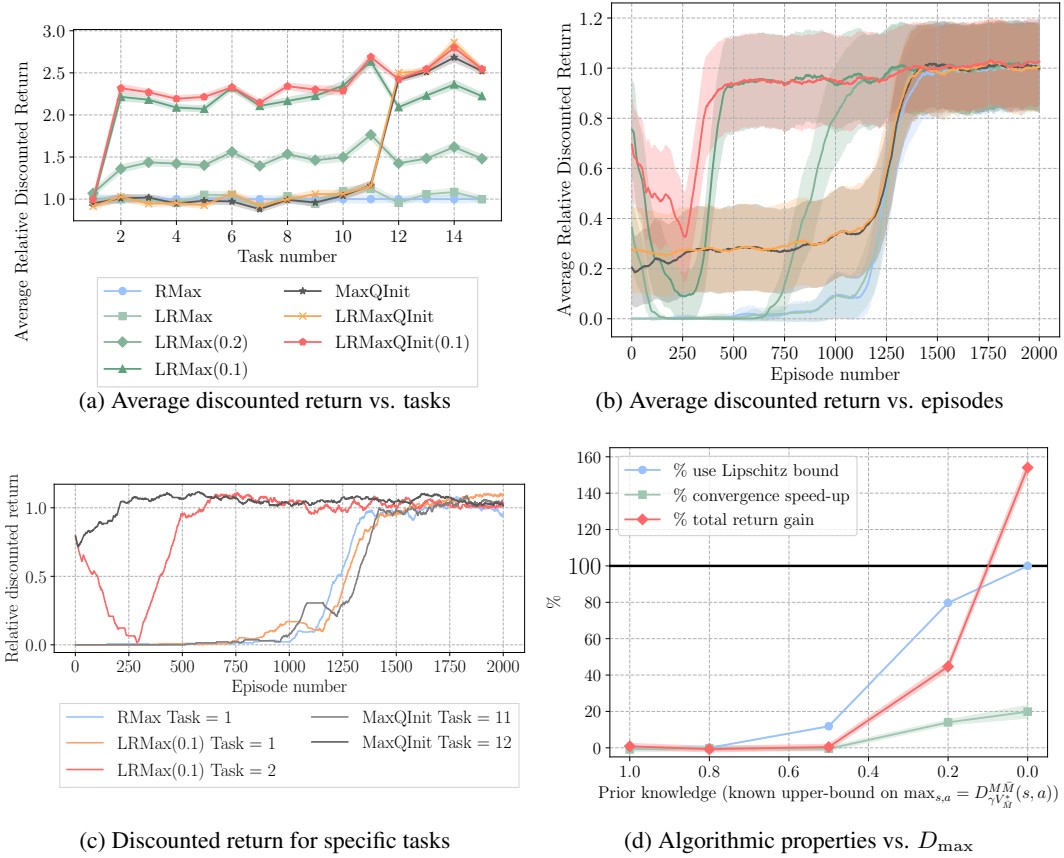

Figure 1: Experimental results

our study with the LRMaxQInit algorithm. Similarly, LRMaxQInit($x$) consists in the latter algorithm, benefiting from prior knowledge $D_{\max} = x$.

The environment we used in all experiments is a variant of the "tight" environment used by Abel et al. (2018). This is a $11 \times 11$ grid-world, the initial state is in the centre, actions are the cardinal moves (Appendix, Section M). The reward is zero everywhere except for the three goal cells in the upper-right corner. Each time a task is sampled, a new reward value is drawn from $[0.8, 1]$ for each of the three goal cells and a probability of slipping (performing a different action than the one selected) is picked in $[0, 0.1]$. Hence, tasks have different reward and transition functions. We sample 15 tasks in sequence among a pool of 5 possible different sampled tasks. Each is run for 2000 episodes of length 10. The operation is repeated 10 times to provide narrow confidence intervals. We used $n_{known} = 10$, $\delta = 0.05$ and $\epsilon = 0.01$ (discussion in Appendix, Section P). We drew tasks from a finite set of five MDPs. This allows the application of MaxQInit and the subsequent comparison below. Note, however, that LRMax does not require the set of MDPs to be finite, which is a noticeable advantage in applicability. Other lifelong RL experiments are reported in the Appendix, Section N.

The results are reported in Figure 1. Figure 1a displays the discounted return for each task, averaged across episodes. Similarly, Figure 1b displays the discounted return for each episode, averaged across tasks (same color code as Figure 1a). Figure 1c displays the discounted return for five specific instances, detailed below. To avoid inter-task disparities, all the aforementioned discounted returns are displayed relatively to an estimator of the optimal expected return for each task. For readability purposes, Figures 1b and 1c display a moving average over 100 episodes. Figure 1d reports the benefits of various values of $D_{\max}$ on the algorithmic properties.

In Figure 1a, we first observe that LRMax benefits from the transfer method, as the average discounted return increases as more tasks are experienced. Moreover, this advantage appears as early as the second task. Conversely, the MaxQInit algorithm needs to wait for task 12 before benefiting from

transfer. As suggested in Section 4.3, various amounts of prior allow the LRMax transfer method to be more or less efficient: a smaller known upper-bound $D_{\max}$ on $\hat{D}^{M\bar{M}}$ causes a larger discounted return gain. Combining both approaches in the LRMaxQInit algorithm outperforms all other methods. Episode-wise, we observe in Figure 1b that the LRMax transfer method allows for faster convergence, hence decreases the sample complexity. Interestingly, LRMax features three stages in the learning process. 1) The first episodes are characterized by a direct exploitation of the transferred knowledge, causing these episodes to yield high payoff. This is due to the combined facts that the Lipschitz bound of Equation 8 is larger on promising regions of $\mathcal{S} \times \mathcal{A}$ seen on previous tasks and the fact that LRMax acts greedily w.r.t. that bound. 2) This high performance regime is followed by the exploration of unknown regions of $\mathcal{S} \times \mathcal{A}$, in our case yielding low returns. Indeed, as promising regions are explored first, the bound becomes tighter for the corresponding state-action pairs, enough for the Lipschitz bound of unknown pairs to become larger, thus driving the exploration towards low payoff regions. Such regions are quickly identified and never revisited thereafter. 3) Eventually, LRMax stops exploring and converges to the optimal policy. Importantly, in all experiments, LRMax never features negative transfer as supported by the provability of the Lipschitz upper-bound with high probability. This is indeed demonstrated by the fact that it is at least as efficient in learning as the no-transfer RMax baseline.

Figure 1c displays the collected returns of RMax, LRMax(0.1), and MaxQInit for specific tasks. We observe that LRMax benefits from the transfer as early as task 2, where the aforementioned 3-stages behavior is visible. Again, MaxQInit needs to wait for task 12 to leverage the transfer method. However, the bound it provides are tight enough to allow for almost zero exploration of the task.

In Figure 1d, we display the following quantities for various values of $D_{\max}$: $\rho_{Lip}$, is the ratio of the time the Lipschitz bound was tighter than the RMax bound $\frac{1}{1-\gamma}$; $\rho_{Speed-up}$, is the relative gain of time steps before convergence when comparing LRMax to RMax. This quantity is estimated based on the last updates of the empirical model $\bar{M}$; $\rho_{Return}$, is the relative total return gain on 2000 episodes of LRMax w.r.t. RMax. First, we observe an increase of $\rho_{Lip}$ as $D_{\max}$ becomes tighter. This means that the Lipschitz bound of Equation 8 becomes effectively smaller than $\frac{1}{1-\gamma}$. This phenomenon leads to faster convergence, indicated by $\rho_{Speed-up}$. Eventually, this increased convergence rate allows for a net total return gain, illustrated by the increase of $\rho_{Return}$.

Overall, in this analysis, we have showed that LRMax benefits from an enhanced sample complexity thanks to the value transfer method. The knowledge of a prior $D_{\max}$ further increases this benefit. The method is comparable to the MaxQInit method and has some advantages such as the early fitness for use and the applicability to infinite sets of tasks. Moreover, the transfer is non-negative while preserving the PAC-MDP guarantees of the algorithm. Additionally to the analysis performed here, we show in the Appendix, Section O that, when provided with any prior knowledge $D_{\max}$, LRMax increasingly stops using this prior as the task is explored. This confirms the claim of section 4.3 that providing $D_{\max}$ enables transfer even if it's value is of little importance.

## 6 CONCLUSION

We have studied theoretically the Lipschitz continuity property of the optimal Q-function in the MDP space. This led to a local Lipschitz continuity result, establishing that the optimal Q-functions of two close MDPs are themselves close to each other. This distance between Q-functions can be computed by Dynamic Programming. We then proposed a value-transfer method using this continuity property with the Lipschitz RMax algorithm, practically implementing this approach in the Lifelong RL setting. The algorithm preserves PAC-MDP guarantees, accelerates the learning in subsequent tasks and performs non-negative transfer. Potential improvements of the algorithm were discussed in the form of prior knowledge introduction on the maximum distance between models and online estimation with high probability of this distance. We showcased the algorithm in lifelong RL experiments and demonstrated empirically its ability to accelerate learning. The results also confirm that no negative transfer occurs, regardless of parameter settings. It should be noted that our approach can directly extend other PAC-MDP algorithms (Szita and Szepesvári, 2010; Rao and Whiteson, 2012; Pazis et al., 2016; Dann et al., 2017) to the Lifelong setting. In hindsight, we believe this contribution provides a sound basis to non-negative value transfer via MDP similarity, a development that was lacking in the literature. Key insights for the practitioner lie both in the theoretical analysis and in the practical derivation of a transfer scheme that achieves non-negative transfer with PAC guarantees.

ACKNOWLEDGEMENTS

Omitted for anonymity.

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

APPENDIX

## A   A NEGATIVE TRANSFER EXAMPLE

In their paper, Song et al. (2016) propose two transfer methods based on the metric between MDPs they introduce, stemming from the bi-simulation metric introduced by Ferns et al. (2004). The intuition is that, for a new target task, the value function of the closest source task in terms of that metric is used as an initialization. However, if no similar source task is available, using the closest task's value function as an initialization can lead to negative transfer. We here understand negative transfer as the fact that it prevents a learning algorithm to converge to the optimal policy while interacting with a new task. We make the hypothesis that the learning algorithm acts greedily w.r.t. the current Q-value function. This is for example the behavior of the RMax algorithm (Brafman and Tennenholtz, 2002). We now illustrate a negative transfer case with an example. Let us consider the 2-states MDP of Figure 2. We assume that the transitions are deterministic and the initial state is

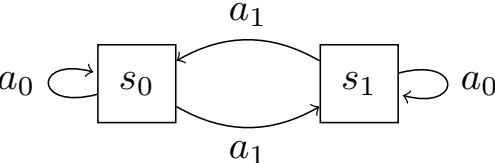

Figure 2: 2-states MDP

always $s_0$. In the first MDP $M_1 \in \mathcal{M}$, the reward is 0 everywhere except for $R_{s_0}^{a_0} = 1$. In the second MDP $M_2 \in \mathcal{M}$, the reward is 0 everywhere except for $R_{s_1}^{a_1} = 1$. With a discount factor $\gamma = 0.9$, the value functions and Q-functions of both MDPs are summarized in Table 3 Using the weighted

|  | $V_{M_1}^*(\cdot)$ | $Q_{M_1}^*(\cdot, a_0)$ | $Q_{M_1}^*(\cdot, a_1)$ | $V_{M_2}^*(\cdot)$ | $Q_{M_2}^*(\cdot, a_0)$ | $Q_{M_2}^*(\cdot, a_1)$ |
|---|---|---|---|---|---|---|
| $s_0$ | 10 | 10 | 8.1 | 4.74 | 4.26 | 4.74 |
| $s_1$ | 9 | 8.1 | 9 | 5.26 | 4.74 | 5.26 |

Figure 3: Value functions and Q-functions of MDPs $M_1$ and $M_2$

transfer technique from $M_1$ to $M_2$ proposed by Song et al. (2016) (Definition 4.1), the Q-function described below is used as an initialization for the exploration of $M_2$.

$$Q_{M_2}^{\text{transfer}}(s_0, a_0) = 2.03$$
$$Q_{M_2}^{\text{transfer}}(s_0, a_1) = 2.25$$
$$Q_{M_2}^{\text{transfer}}(s_1, a_0) = 2.5$$
$$Q_{M_2}^{\text{transfer}}(s_1, a_1) = 2.03$$

First, $Q_{M_2}^{\text{transfer}}$ does not respect the principle of "optimism under the face of uncertainty" that often results in sound and efficient exploration (Strehl et al., 2009; Brafman and Tennenholtz, 2002; Sutton and Barto, 1998). Further, a greedy policy w.r.t. $Q_{M_2}^{\text{transfer}}$ would never discover the state-action pair $s_1, a_1$ in $M_2$ which is the maximum-reward pair. Instead, the agent would go from $s_0$ to $s_1$ and perform self-loops thereafter.

As a conclusion, this negative transfer example motivates the need for distance between MDPs not only to account for the best-source task to use for transfer but also to discourage the transfer when the distance is too high. The approach we develop in this paper used the distance to build optimistic upper-bounds on the Q-function. Those upper-bounds are simply of no use when the distance is too high which is equivalent as avoiding transfer.

## B  Discussion on metrics and related notions

A metric on a set $X$ is a function $m : X \times X \to \mathbb{R}$ which has the following properties for any $x, y, z \in X$:

1. $m(x, y) \geq 0$,
2. $m(x, y) = 0 \Leftrightarrow x = y$,
3. $m(x, y) = m(y, x)$,
4. $m(x, z) \leq m(x, y) + m(y, z)$.

With only $m(x, x) = 0$ instead of property 2, $m$ would be a *pseudo-metric*. Without property 3, one has a *quasi-metric*. Without property 3 and 4, and when $X$ is a set of probability measures, one has a *divergence*.

In Definition 1, $D_{M,f}^{\bar{M}}(s, a)$ is indeed a pseudo-metric over MDPs since the choice of $f$ can lead to a zero distance between different models.

The local MDP dissimilarity between MDPs $d_M^{\bar{M}}(s, a)$ of Proposition 1 does not respect properties 2 and 3, hence the name *dissimilarity*. The $\Delta_M^{\bar{M}}(s, a) \triangleq \min\left\{d_M^{\bar{M}}(s, a), d_{\bar{M}}^{M}(s, a)\right\}$ quantity, however, regains property 3 and is hence a pseudo-metric. An important consequence is that Proposition 1 is "in the spirit" of a Lipschitz continuity theorem but cannot be called as such, hence the name *pseudo-Lipschitz continuity*.

The same goes for the global dissimilarity $d_M^{\bar{M}} = \frac{1}{1-\gamma} \max_{s,a \in \mathcal{S} \times \mathcal{A}} \left[ D_{M, \gamma V_{\bar{M}}^*}^{\bar{M}}(s, a) \right]$. However, using $\min\left\{d_M^{\bar{M}}, d_{\bar{M}}^{M}\right\}$ allows to regain property 3 and makes this quantity a pseudo-metric again between MDPs.

## C  Proof of Proposition 1

**Lemma 1.** *Given two MDPs $M$ and $\bar{M}$, this equation on $d$ is a fixed-point equation admitting a unique solution which we call $d_M^{\bar{M}}$*

$$d(s, a) = D_{M, \gamma V_{\bar{M}}^*}^{\bar{M}}(s, a) + \gamma \sum_{s'} T_{ss'}^a \max_{a'} d(s', a'), \forall s, a \in \mathcal{S} \times \mathcal{A}.$$

*Proof of Lemma 1.* The proof follows closely that in Puterman (2014) that proves that the Bellman operator over value functions is a contraction mapping. Let $d_1$ and $d_2$ be two functions from $\mathcal{S} \times \mathcal{A}$ to $\mathbb{R}$ and let $L$ be the functional operator that maps any function $d : \mathcal{S} \times \mathcal{A} \to \mathbb{R}$ to

$$Ld : s, a \mapsto D_{M, \gamma V_{\bar{M}}^*}^{\bar{M}}(s, a) + \gamma \sum_{s'} T_{ss'}^a \max_{a'} d(s', a').$$

Then $Ld_1(s, a) - Ld_2(s, a) = \gamma \sum_{s'} T_{ss'}^a [\max_{a'} d_1(s', a') - \max_{a'} d_2(s', a')]$. But $\max_{a'} d_1(s', a') - \max_{a'} d_2(s', a') \leq \max_{a'} [d_1(s', a') - d_2(s', a')] \leq \|d_1 - d_2\|_\infty$. And so $\|Ld_1 - Ld_2\|_\infty \leq \gamma \|d_1 - d_2\|_\infty$. Since $\gamma < 1$, $L$ is a contraction mapping in the metric space $(\mathcal{S} \times \mathcal{A}, \|\cdot\|_\infty)$. This metric space being complete and non-empty, it follows from Banach fixed point theorem that $d = Ld$ admits a single solution. $\square$

Lemma 1 guarantees the existence of $d_M^{\bar{M}}$. Proposition 1 states that for any two MDPs $M$ and $\bar{M}$ and for all $(s, a) \in \mathcal{S} \times \mathcal{A}$, $|Q_M^*(s, a) - Q_{\bar{M}}^*(s, a)| \leq \min\left\{d_M^{\bar{M}}(s, a), d_{\bar{M}}^{M}(s, a)\right\}$.

*Proof of Proposition 1.* The proof is by induction. The Value Iteration sequence of iterates $(Q_M^n)_{n \in \mathbb{N}}$ for task $M$ is:

$$Q_M^0(s, a) = 0, \forall s, a \in \mathcal{S} \times \mathcal{A}$$

$$Q_M^{n+1}(s, a) = R_s^a + \gamma \sum_{s' \in \mathcal{S}} T_{ss'}^a \max_{a' \in \mathcal{A}} Q_M^n(s', a'), \forall s, a \in \mathcal{S} \times \mathcal{A}.$$

It is obvious that $Q_M^0(s,a) - Q_{\bar{M}}^0(s,a) \le d_M^{\bar{M}}(s,a)$. Suppose that $|Q_M^n(s,a) - Q_{\bar{M}}^n(s,a)| \le d_M^{\bar{M}}(s,a)$. Then:

$$
\begin{aligned}
\left|Q_M^{n+1}(s,a) - Q_{\bar{M}}^{n+1}(s,a)\right| &= \left|R_s^a - \bar{R}_s^a + \gamma \sum_{s'\in\mathcal{S}} \left[ T_{ss'}^a \max_{a'\in\mathcal{A}} Q_M^n(s',a') - \bar{T}_{ss'}^a \max_{a'\in\mathcal{A}} Q_{\bar{M}}^n(s',a') \right]\right| \\
&\le \left|R_s^a - \bar{R}_s^a\right| + \gamma \sum_{s'\in\mathcal{S}} \left| T_{ss'}^a \max_{a'\in\mathcal{A}} Q_M^n(s',a') - \bar{T}_{ss'}^a \max_{a'\in\mathcal{A}} Q_{\bar{M}}^n(s',a') \right| \\
&\le \left|R_s^a - \bar{R}_s^a\right| + \gamma \sum_{s'\in\mathcal{S}} \max_{a'\in\mathcal{A}} Q_{\bar{M}}^n(s',a') \left|T_{ss'}^a - \bar{T}_{ss'}^a\right| \\
&\quad + \gamma \sum_{s'\in\mathcal{S}} T_{ss'}^a \left| \max_{a'\in\mathcal{A}} Q_M^n(s',a') - \max_{a'\in\mathcal{A}} Q_{\bar{M}}^n(s',a') \right| \\
&\le \left|R_s^a - \bar{R}_s^a\right| + \sum_{s'\in\mathcal{S}} \gamma V_{\bar{M}}^*(s') \left|T_{ss'}^a - \bar{T}_{ss'}^a\right| \\
&\quad + \gamma \sum_{s'\in\mathcal{S}} T_{ss'}^a \max_{a'\in\mathcal{A}} \left|Q_M^n(s',a') - Q_{\bar{M}}^n(s',a')\right| \\
&\le D_{M,\gamma V_{\bar{M}}^*}^{\bar{M}}(s,a) + \gamma \sum_{s'\in\mathcal{S}} T_{ss'}^a \max_{a'} d_M^{\bar{M}}(s',a')
\end{aligned}
$$

Since $Q_M^*$ and $Q_{\bar{M}}^*$ are respectively the limits of the $(Q_M^n)_{n\in\mathbb{N}}$ and $(Q_{\bar{M}}^n)_{n\in\mathbb{N}}$ sequences, the result that $|Q_M^*(s,a) - Q_{\bar{M}}^*(s,a)| \le d_M^{\bar{M}}(s,a)$ follows from passage to the limit.

By symmetry, on also has $|Q_M^*(s,a) - Q_{\bar{M}}^*(s,a)| \le d_{\bar{M}}^M(s,a)$ and thus $|Q_M^*(s,a) - Q_{\bar{M}}^*(s,a)| \le \min\left\{ d_M^{\bar{M}}(s,a), d_{\bar{M}}^M(s,a) \right\}$. □

## D    SIMILAR RESULTS TO PROPOSITION 1

Similar results to Proposition 1 can be derived with a similar proof as in Section C. The first result is for the value function and is stated below.

**Proposition** (Local bound on the distance between value functions). *For any two MDPs $M$ and $\bar{M}$, for all $s \in \mathcal{S}$,*

$$
|V_M^*(s) - V_{\bar{M}}^*(s)| \le \max_{a\in\mathcal{A}} \Delta_M^{\bar{M}}(s,a)
$$

*where the local MDP pseudo-metric $\Delta_M^{\bar{M}}(s,a)$ has the same definition as in Proposition 1.*

Another result can be derived for any policy $\pi$ that one wishes to evaluate in both MDPs. For the sake of generality, we state the result for any stochastic policy mapping states to distributions over actions. A deterministic policy is a stochastic policy choosing the selected action with probability 1 and the others with probability 0.

**Proposition** (Local bound on the distance between value and Q-value functions for any policy.). *For any two MDPs $M$ and $\bar{M}$, for a stochastic policy $\pi$, for all $s,a \in \mathcal{S} \times \mathcal{A}$,*

$$
|V_M^\pi(s) - V_{\bar{M}}^\pi(s)| \le \Delta_M^{\pi,\bar{M}}(s)
$$

*where $d_M^{\pi,\bar{M}}(s)$ is defined with the following fixed-point equation:*

$$
d_M^{\pi,\bar{M}}(s) = \mathbb{E}_{a\sim\pi} \left[ D_{M,\gamma V_{\bar{M}}^*}^{\bar{M}}(s,a) + \gamma \sum_{s'\in\mathcal{S}} T_{ss'}^a d_M^{\pi,\bar{M}}(s') \right],
$$

*and $\Delta_M^{\pi,\bar{M}}(s) = \min\left\{ d_M^{\pi,\bar{M}}(s), d_{\bar{M}}^{\pi,M}(s) \right\}$.*

# E GLOBAL PSEUDO-LIPSCHITZ CONTINUITY RESULT

A consequence of Proposition 1 is a global pseudo-Lipschitz continuity property:

**Proposition 8** (Global pseudo-Lipschitz continuity). *For two MDPs $M$, $\bar{M}$, for all $(s, a) \in \mathcal{S} \times \mathcal{A}$,*

$$|Q_M^*(s,a) - Q_{\bar{M}}^*(s,a)| \leq \min\left\{\delta_M^{\bar{M}}, \delta_{\bar{M}}^M\right\}, \text{ with } \delta_M^{\bar{M}} \triangleq \frac{1}{1-\gamma} \max_{s,a \in \mathcal{S} \times \mathcal{A}} \left\{D_{\gamma V_{\bar{M}}^*}^{M\bar{M}}(s,a)\right\}. \quad (10)$$

Despite being interesting from a theoretical perspective, we do not use this result for transfer because it is impractical to compute. Indeed, estimating the maximum in Equation 10 might be as hard as solving both MDPs (which, when it happens, is too late for transfer to be useful).

*Proof.* The proof is by induction and reuses the notations introduced in the proof of Proposition 1. It is immediate that

$$\begin{aligned} \left|Q_M^0(s,a) - Q_{\bar{M}}^0(s,a)\right| &\leq d_M^{\bar{M}}, \text{ and} \\ \left|Q_M^0(s,a) - Q_{\bar{M}}^0(s,a)\right| &\leq d_{\bar{M}}^M. \end{aligned}$$

Hence, the result holds for $n = 0$. Let us suppose that

$$\begin{aligned} \left|Q_M^n(s,a) - Q_{\bar{M}}^n(s,a)\right| &\leq d_M^{\bar{M}}, \text{ and} \\ \left|Q_M^n(s,a) - Q_{\bar{M}}^n(s,a)\right| &\leq d_{\bar{M}}^M. \end{aligned}$$

Then,

$$\begin{aligned} \left|Q_M^{n+1}(s,a) - Q_{\bar{M}}^{n+1}(s,a)\right| &\leq D_{M,\gamma V_{\bar{M}}^*}^{\bar{M}}(s,a) + \gamma \sum_{s' \in \mathcal{S}} T_{ss'}^a \max_{a' \in \mathcal{A}} \left|Q_M^n(s',a') - Q_{\bar{M}}^n(s',a')\right| \\ &\leq \max_{s,a \in \mathcal{S} \times \mathcal{A}} \left[D_{M,\gamma V_{\bar{M}}^*}^{\bar{M}}(s,a)\right] + \gamma \sum_{s' \in \mathcal{S}} T_{ss'}^a \frac{1}{1-\gamma} \max_{s,a \in \mathcal{S} \times \mathcal{A}} \left[D_{M,\gamma V_{\bar{M}}^*}^{\bar{M}}(s,a)\right] \\ &\leq \max_{s,a \in \mathcal{S} \times \mathcal{A}} \left[D_{M,\gamma V_{\bar{M}}^*}^{\bar{M}}(s,a)\right] \left(1 + \frac{\gamma}{1-\gamma}\right) \\ &\leq d_M^{\bar{M}} \end{aligned}$$

$\square$

# F PROOF OF PROPOSITION 2

*Proof.* The result is clear for all $s, a \notin K$ since the Lipschitz bounds are provably greater than $Q_M^*$. For $s, a \in K$, the result is by induction. Let us consider the Dynamic Programming (Bellman, 1957) sequences converging to $Q_M^*$ and $U$ at rank $n$ whose definitions follow:

$$\begin{cases} Q_{M,0}^*(s,a) = 0 \\ Q_{M,n}^*(s,a) = R_s^a + \gamma \sum_{s'} T_{ss'}^a \max_{a'} Q_{M,n-1}^*(s',a') \end{cases}$$

$$\begin{cases} U_0(s,a) = 0 \\ U_n(s,a) = R_s^a + \gamma \sum_{s'} T_{ss'}^a \max_{a'} U_{n-1}(s',a') \end{cases}$$

Obviously, $Q_{M,0}^*(s,a) \leq U_0(s,a)$. Suppose the property true at rank $n$ and consider rank $n + 1$:

$$\begin{aligned} Q_{M,n+1}^*(s,a) - U_{n+1}(s,a) &= \gamma \sum_{s'} T_{ss'}^a \left(\max_{a'} Q_{M,n}^*(s',a') - \max_{a'} U_n(s',a')\right) \\ &\leq \gamma \sum_{s'} T_{ss'}^a \max_{a'} \left(Q_{M,n}^*(s',a') - U_n(s',a')\right) \\ &\leq 0 \end{aligned}$$

Which concludes the proof by induction. The result holds by passage to the limit since the considered Dynamic Programming sequences converge to the true functions. $\square$

## G PROOF OF PROPOSITION 3

Consider two tasks $M = \langle T, R \rangle$ and $\bar{M} = \langle \bar{T}, \bar{R} \rangle$, with $K$ and $\bar{K}$ the respective sets of state-action pairs where their learned models $\hat{M} = \langle \hat{T}, \hat{R} \rangle$ and $\hat{\bar{M}} = \langle \hat{\bar{T}}, \hat{\bar{R}} \rangle$ are known with accuracy $\epsilon$ in $\mathcal{L}_1$-norm with probability at least $1 - \delta$, i.e. we have that,

$$\mathbf{Pr}\left(|R_s^a - \hat{R}_s^a| \le \epsilon\right) \ge 1 - \delta, \forall s, a \in K, \tag{11}$$

$$\mathbf{Pr}\left(\|T_{ss'}^a - \hat{T}_{ss'}^a\|_1 \le \epsilon\right) \ge 1 - \delta, \forall s, a \in K, \tag{12}$$

and the same goes for $\bar{M}$ and its learned model $\hat{\bar{M}}$. We state the result for each one of the three cases 1) $s, a \in K \cap \bar{K}$, 2) $s, a \in K \cap \bar{K}^c$ and 3) $s, a \in K^c \cap \bar{K}^c$, the case $s, a \in K^c \cap \bar{K}$ being the symmetric of case 2).

1) If $s, a \in K \cap \bar{K}$, then properties 11 and 12 hold for both $\langle R, T \rangle$ with $\langle \hat{R}, \hat{T} \rangle$ and $\langle \bar{R}, \bar{T} \rangle$ with $\langle \hat{\bar{R}}, \hat{\bar{T}} \rangle$. We have by definition:

$$D_{\gamma V_{\bar{M}}^*}^{M \bar{M}}(s, a) = |R_s^a - \bar{R}_s^a| + \gamma \sum_{s' \in \mathcal{S}} V_{\bar{M}}^*(s')|T_{ss'}^a - \bar{T}_{ss'}^a|. \tag{13}$$

The first term of the RHS of Equation 13 respects the following sequence of inequalities with probability at least $1 - \delta$:

$$|R_s^a - \bar{R}_s^a| \le |R_s^a - \hat{R}_s^a| + |\hat{R}_s^a - \hat{\bar{R}}_s^a| + |\bar{R}_s^a - \hat{\bar{R}}_s^a|$$
$$\le |\hat{R}_s^a - \hat{\bar{R}}_s^a| + 2\epsilon. \tag{14}$$

The second term of the RHS of Equation 13 respects the following sequence of inequalities with probability at least $1 - \delta$:

$$\gamma \sum_{s' \in \mathcal{S}} V_{\bar{M}}^*(s')|T_{ss'}^a - \bar{T}_{ss'}^a| \le \gamma \sum_{s' \in \mathcal{S}} \bar{V}(s')\left(|T_{ss'}^a - \hat{T}_{ss'}^a| + |\hat{T}_{ss'}^a - \hat{\bar{T}}_{ss'}^a| + |\bar{T}_{ss'}^a - \hat{\bar{T}}_{ss'}^a|\right)$$
$$\le \gamma \max_{s'} \bar{V}(s') \sum_{s' \in \mathcal{S}} |T_{ss'}^a - \hat{T}_{ss'}^a| + \gamma \sum_{s' \in \mathcal{S}} \bar{V}(s')|\hat{T}_{ss'}^a - \hat{\bar{T}}_{ss'}^a| +$$
$$\gamma \max_{s'} \bar{V}(s') \sum_{s' \in \mathcal{S}} |\bar{T}_{ss'}^a - \hat{\bar{T}}_{ss'}^a|$$
$$\le \gamma \sum_{s' \in \mathcal{S}} \bar{V}(s')|\hat{T}_{ss'}^a - \hat{\bar{T}}_{ss'}^a| + 2\epsilon\gamma \max_{s'} \bar{V}(s'). \tag{15}$$

Summation of Equations 14 and 15 reveals $\hat{D}^{M\bar{M}}(s, a) = |\hat{R}_s^a - \hat{\bar{R}}_s^a| + \gamma \sum_{s' \in \mathcal{S}} \bar{V}(s')|\hat{T}_{ss'}^a - \hat{\bar{T}}_{ss'}^a|$ on the RHS of the inequality. Remarking this, we can upper-bound the model pseudo-distance of Equation 13 by the expected quantity with probability at least $1 - \delta$, proving the Proposition for case 1):

$$D_{\gamma V_{\bar{M}}^*}^{M \bar{M}}(s, a) \le \hat{D}^{M\bar{M}}(s, a) + 2\epsilon\left(1 + \gamma \max_{s'} \bar{V}(s')\right).$$

2) If $s, a \in K \cap \bar{K}^c$, then properties 11 and 12 hold for $\langle R, T \rangle$ with $\langle \hat{R}, \hat{T} \rangle$ only. Similarly to the proof of case 1), we upper bound sequentially the two terms of the RHS of Equation 13. With probability at least $1 - \delta$, we have the following:

$$|R_s^a - \bar{R}_s^a| \le |R_s^a - \hat{R}_s^a| + |\hat{R}_s^a - \bar{R}_s^a|$$
$$\le \epsilon + \max_{\bar{R}} |\hat{R}_s^a - \bar{R}|. \tag{16}$$

Similarly, with probability at least $1 - \delta$, we have:

$$\gamma \sum_{s' \in \mathcal{S}} V_{\bar{M}}^*(s')|T_{ss'}^a - \bar{T}_{ss'}^a| \le \gamma \sum_{s' \in \mathcal{S}} \bar{V}(s')\left(|T_{ss'}^a - \hat{T}_{ss'}^a| + |\hat{T}_{ss'}^a - \bar{T}_{ss'}^a|\right)$$
$$\le \gamma \max_{s'} \bar{V}(s')\epsilon + \gamma \max_{\bar{T}} \sum_{s' \in \mathcal{S}} \bar{V}(s')|\hat{T}_{ss'}^a - \bar{T}_{s'}|. \tag{17}$$

Combining inequalities 16 and 17, we get the following with probability at least $1 - \delta$, noticing $D_{\gamma V_{\bar{M}}^*}^{M\bar{M}}(s, a)$ on the LHS:

$$D_{\gamma V_{\bar{M}}^*}^{M\bar{M}}(s, a) \leq \max_{\bar{\mu} \in \mathcal{M}} D_{\gamma \bar{V}}^{\hat{M}\bar{\mu}}(s, a) + \epsilon \left(1 + \gamma \max_{s'} \bar{V}(s')\right),$$

which is the expected result.

3) If $s, a \in K^c \cap \bar{K}^c$, then properties 11 and 12 do not hold. In such a case, the result

$$D_{\gamma V_{\bar{M}}^*}^{M\bar{M}}(s, a) \leq \max_{\mu, \bar{\mu} \in \mathcal{M}^2} D_{\gamma \bar{V}}^{\mu\bar{\mu}}(s, a)$$

is straightforward by remarking that $V_{\bar{M}}^*(s) \leq \bar{V}(s)$ with probability at least $1 - \delta$.

# H ANALYTICAL CALCULATION OF $\hat{D}^{M\bar{M}}$ IN PROPOSITION 3

Consider two tasks $M = \langle T, R \rangle$ and $\bar{M} = \langle \bar{T}, \bar{R} \rangle$, with $K$ and $\bar{K}$ the respective sets of state-action pairs where their learned models $\hat{M} = \langle \hat{T}, \hat{R} \rangle$ and $\hat{\bar{M}} = \langle \hat{\bar{T}}, \hat{\bar{R}} \rangle$ are known with accuracy $\epsilon$ in $\mathcal{L}_1$-norm with probability at least $1 - \delta$. We note $V_{\max}$, a known upper-bound on the maximum achievable value. In the worst case where one does not have any information on the value of $V_{\max}$, one can always set $V_{\max} = \frac{1}{1-\gamma}$. We recall the definition of the upper bound on the pseudo-metric between models:

$$\hat{D}^{M\bar{M}}(s, a) = \begin{cases} D_{\gamma \bar{V}}^{\hat{M}\hat{\bar{M}}}(s, a) + 2B & \text{if } (s, a) \in K \cap \bar{K}, \\ \max_{\bar{\mu} \in \mathcal{M}} D_{\gamma \bar{V}}^{\hat{M}\bar{\mu}}(s, a) + B & \text{if } (s, a) \in K \cap \bar{K}^c, \\ \max_{\mu \in \mathcal{M}} D_{\gamma \bar{V}}^{\mu\hat{\bar{M}}}(s, a) + B & \text{if } (s, a) \in K^c \cap \bar{K}, \\ \max_{\mu, \bar{\mu} \in \mathcal{M}^2} D_{\gamma \bar{V}}^{\mu\bar{\mu}}(s, a) & \text{if } (s, a) \in K^c \cap \bar{K}^c. \end{cases} \tag{18}$$

with $B = \epsilon \left(1 + \gamma \max_{s'} \bar{V}(s')\right)$ and $D_f^{M\bar{M}}$ defined as in Equation 13. We detail the computation of $\hat{D}^{M\bar{M}}(s, a)$ for each cases 1) $s, a \in K \cap \bar{K}$, 2) $s, a \in K \cap \bar{K}^c$ (the $s, a \in K^c \cap \bar{K}$ is symmetric to this one), and 3) $s, a \in K^c \cap \bar{K}^c$. Recall that we consider a finite, countable, state-action space $\mathcal{S} \times \mathcal{A}$.

1) If $s, a \in K \cap \bar{K}$, we have

$$\hat{D}^{M\bar{M}}(s, a) = D_{\gamma \bar{V}}^{\hat{M}\hat{\bar{M}}}(s, a) + 2B$$
$$= |\hat{R}_s^a - \hat{\bar{R}}_s^a| + \gamma \sum_{s' \in \mathcal{S}} \bar{V}(s') |\hat{T}_{ss'}^a - \hat{\bar{T}}_{ss'}^a| + 2\epsilon \left(1 + \gamma \max_{s'} \bar{V}(s')\right).$$

Since $s, a$ is a known state-action pair, everything is known and computable in this last equation. Note that $\max_{s'} \bar{V}(s')$ can be tracked along the updates of $\bar{V}$ and thus its computation does not induce any additional complexity.

2) If $s, a \in K \cap \bar{K}^c$, we have

$$\hat{D}^{M\bar{M}}(s, a) = \max_{\bar{\mu} \in \mathcal{M}} D_{\gamma \bar{V}}^{\hat{M}\bar{\mu}}(s, a) + B$$

$$= \max_{\bar{R}_s^a, \bar{T}_{ss'}^a} \left(|\hat{R}_s^a - \bar{R}_s^a| + \gamma \sum_{s' \in \mathcal{S}} \bar{V}(s') |\hat{T}_{ss'}^a - \bar{T}_{ss'}^a|\right) + \epsilon \left(1 + \gamma \max_{s'} \bar{V}(s')\right),$$

$$= \max_{r \in [0,1]} |\hat{R}_s^a - r| + \gamma \max_{\substack{t \in [0,1]^{|\mathcal{S}|} \\ \sum t = 1}} \left(\sum_{s' \in \mathcal{S}} \bar{V}(s') |\hat{T}_{ss'}^a - t_{s'}|\right) + \epsilon \left(1 + \gamma \max_{s'} \bar{V}(s')\right).$$

First, we have

$$\max_{r \in [0,1]} |\hat{R}_s^a - r| = \max\left\{\hat{R}_s^a, 1 - \hat{R}_s^a\right\}.$$

Maximizing the $\max_{t \in [0,1]^{|\mathcal{S}|}}$ term is maximizing a convex combination of $\bar{V}$ (whose values are all positive) whose terms are not independent (since the $t_{s'}$ terms should sum to one). This is easily cast as a linear programming problem. A straightforward (simplex-like) resolution procedure consists in progressively adding mass on the terms that will maximize the convex combination as follows:

- $t_{s'} = 0, \forall s' \in \mathcal{S}$
- $l$ = Sort states by decreasing value of $\bar{V}$
- While $\sum_{s \in \mathcal{S}} t(s) < 1$
  - $s'$ = pop first state in $l$
  - Assign $t(s') \leftarrow \arg\max_{t \in [0,1]} |\hat{T}_{ss'}^a - t|$ to $s'$ (note that $t_{s'} \in \{0,1\}$)
  - If $\sum_{s \in \mathcal{S}} t_s > 1$, then $t_{s'} \leftarrow 1 - \sum_{s \in \mathcal{S} \setminus s'} t(s)$

This allows calculating the maximum over transition models.

There is however a simpler computation that almost always yields the same result (when it does not, it provides an upper bound) and does not require the burden of the previous procedure. Consider the subset of states for which $\bar{V}(s') = \max_s \bar{V}(s)$ (often these are states in $\bar{K}^c$). Among those states, let us suppose there exists $s^+$ unreachable from $s, a$, according to $\hat{T}$, that is $\hat{T}_{ss^+}^a = 0$. If $\bar{M}$ has not been fully explored, as is often the case in RMax, there may be many such states. Then the distribution $t$ with all its mass on $s^+$ is a maximizer of the $\max_{t \in [0,1]^{|\mathcal{S}|}}$ term. Conversely, if such a state does not exist (that is, if for all such states $\hat{T}_{ss^+}^a > 0$), then $\max_s \bar{V}(s)$ is an upper bound on the $\max_{t \in [0,1]^{|\mathcal{S}|}}$ term. Therefore:

$$\max_{t \in [0,1]^{|\mathcal{S}|}} \left( \sum_{s' \in \mathcal{S}} \bar{V}(s') |\hat{T}_{ss'}^a - t_{s'}| \right) \leq \max_s \bar{V}(s), \text{ with equality in many cases.}$$

3) If $s, a \in K^c \cap \bar{K}^c$, the resolution is trivial and we have

$$\hat{D}^{M\bar{M}}(s,a) = \max_{\mu, \bar{\mu} \in \mathcal{M}^2} D_{\gamma \bar{V}}^{\mu \bar{\mu}}(s,a)$$

$$= \max_{R_s^a, T_{ss'}^a, \bar{R}_s^a, \bar{T}_{ss'}^a} \left( |R_s^a - \bar{R}_s^a| + \gamma \sum_{s' \in \mathcal{S}} \bar{V}(s') |T_{ss'}^a - \bar{T}_{ss'}^a| \right)$$

$$= \max_{r, \bar{r} \in [0,1]} |r - \bar{r}| + \gamma \max_{\substack{t, \bar{t} \in [0,1]^{|\mathcal{S}|} \\ \sum t = 1 \\ \sum \bar{t} = 1}} \sum_{s' \in \mathcal{S}} \bar{V}(s') |t_{s'} - \bar{t}_{s'}|$$

$$= 1 + \gamma \max_s \bar{V}(s).$$

# I  PROOF OF PROPOSITION 4

**Lemma 2.** *Given two tasks $M$ and $\bar{M}$, $K$ the set of state-action pairs for which $\langle R, T \rangle$ is known with accuracy $\epsilon$ in $\mathcal{L}_1$-norm with probability at least $1 - \delta$. If $\gamma(1 + \epsilon) < 1$, this equation on $\hat{d}$ is a fixed-point equation admitting a unique solution which we call $\hat{d}_M^{\bar{M}}$*

$$\hat{d}(s,a) = \begin{cases} \hat{D}^{M\bar{M}}(s,a) + \gamma \left( \sum_{s' \in \mathcal{S}} \hat{T}_{ss'}^a \max_{a' \in \mathcal{A}} \hat{d}(s',a') + \epsilon \max_{s',a' \in \mathcal{S} \times \mathcal{A}} \hat{d}(s',a') \right) & \text{if } s, a \in K, \\ \hat{D}^{M\bar{M}}(s,a) + \gamma \max_{s',a' \in \mathcal{S} \times \mathcal{A}} \hat{d}(s',a') & \text{otherwise.} \end{cases}$$

*Proof of Lemma 2.* The proof is similar to the proof of Lemma 1. Let $d_1$ and $d_2$ be two functions from $\mathcal{S} \times \mathcal{A}$ to $\mathbb{R}$ and let $L$ be the functional operator that maps any function $d : \mathcal{S} \times \mathcal{A} \to \mathbb{R}$ to

$$Ld : s, a \mapsto \begin{cases} \hat{D}^{M\bar{M}}(s,a) + \gamma \left( \sum_{s' \in \mathcal{S}} \hat{T}_{ss'}^a \max_{a' \in \mathcal{A}} d(s',a') + \epsilon \max_{s',a' \in \mathcal{S} \times \mathcal{A}} d(s',a') \right) & \text{if } s, a \in K, \\ \hat{D}^{M\bar{M}}(s,a) + \gamma \max_{s',a' \in \mathcal{S} \times \mathcal{A}} d(s',a') & \text{otherwise.} \end{cases}$$

If $s, a \in K$, we have

$$
Ld_1(s,a) - Ld_2(s,a) = \gamma \sum_{s'} T^a_{ss'} \left( \max_{a'} d_1(s',a') - \max_{a'} d_2(s',a') \right) +
$$

$$
\gamma \epsilon \left( \max_{s',a'} d_1(s',a') - \max_{s',a'} d_2(s',a') \right)
$$

$$
\leq (\gamma + \gamma \epsilon) \left( \max_{s',a'} d_1(s',a') - \max_{s',a'} d_2(s',a') \right)
$$

$$
\leq \gamma(1+\epsilon) \max_{s',a'} (d_1(s',a') - d_2(s',a'))
$$

$$
\leq \gamma(1+\epsilon)\|d_1 - d_2\|_\infty.
$$

If $s, a \notin K$, we have

$$
Ld_1(s,a) - Ld_2(s,a) = \gamma \left( \max_{s',a'} d_1(s',a') - \max_{s',a'} d_2(s',a') \right)
$$

$$
\leq \gamma \max_{s',a'} (d_1(s',a') - d_2(s',a'))
$$

$$
= \gamma(1+\epsilon)\|d_1 - d_2\|_\infty.
$$

In both cases, $\|Ld_1 - Ld_2\|_\infty \leq \gamma(1+\epsilon)\|d_1 - d_2\|_\infty$. If $\gamma(1+\epsilon) < 1$, $L$ is a contraction mapping in the metric space $(\mathcal{S} \times \mathcal{A}, \|\cdot\|_\infty)$. This metric space being complete and non-empty, it follows from Banach fixed point theorem that $d = Ld$ admits a single solution. □

*Proof of Proposition 4.* The proof is done by induction, by calculating the values of $d^{\bar{M}}_M$ and $\hat{d}^{\bar{M}}_M$ following the value iteration algorithm. Those values can respectively be computed via the sequences of iterates $(d^n)_{n \in \mathbb{N}}$ and $(\hat{d}^n)_{n \in \mathbb{N}}$ defined as follows:

$$
d^0(s,a) = 0, \forall s, a \in \mathcal{S} \times \mathcal{A}
$$

$$
d^{n+1}(s,a) = D^{M\bar{M}}_{\gamma V^*_{\bar{M}}}(s,a) + \gamma \sum_{s' \in \mathcal{S}} T^a_{ss'} \max_{a' \in \mathcal{A}} d^n(s',a')
$$

and,

$$
\hat{d}^0(s,a) = 0, \forall s, a \in \mathcal{S} \times \mathcal{A},
$$

$$
\hat{d}^{n+1}(s,a) = \begin{cases} \hat{D}^{M\bar{M}}(s,a) + \gamma \left( \sum_{s' \in \mathcal{S}} \hat{T}^a_{ss'} \max_{a' \in \mathcal{A}} \hat{d}^n(s',a') + \epsilon \max_{s',a' \in \mathcal{S} \times \mathcal{A}} \hat{d}^n(s',a') \right) & \text{if } s, a \in K, \\ \hat{D}^{M\bar{M}}(s,a) + \gamma \max_{s',a' \in \mathcal{S} \times \mathcal{A}} \hat{d}^n(s',a') & \text{otherwise.} \end{cases}
$$

The proof at rank $n = 0$ is trivial. Let us assume the proposition $d^n \leq \hat{d}^n, \forall s, a \in \mathcal{S} \times \mathcal{A}$ true at rank $n$ and consider rank $n + 1$. There are two cases, depending on the fact that $s, a$ is in $K$ or not.

If $s, a \in K$, we have

$$
d^{n+1}(s,a) - \hat{d}^{n+1}(s,a) = D^{M\bar{M}}_{\gamma V^*_{\bar{M}}}(s,a) - \hat{D}^{M\bar{M}}(s,a) +
$$

$$
\gamma \sum_{s' \in \mathcal{S}} \left( T^a_{ss'} \max_{a' \in \mathcal{A}} d^n(s',a') - \hat{T}^a_{ss'} \max_{a' \in \mathcal{A}} \hat{d}^n(s',a') \right) +
$$

$$
- \gamma \epsilon \max_{s',a' \in \mathcal{S} \times \mathcal{A}} \hat{d}^n(s',a').
$$

Using Proposition 3, we have that $\hat{D}^{M\bar{M}}(s,a)$ is an upper bound on $D^{M\bar{M}}_{\gamma V^*_{\bar{M}}}(s,a)$ with probability at least $1 - \delta$. Hence

$$
\mathbf{Pr}\left( D^{M\bar{M}}_{\gamma V^*_{\bar{M}}}(s,a) - \hat{D}^{M\bar{M}}(s,a) \leq 0 \right) \geq 1 - \delta.
$$

This plus the fact that $d^n \leq \hat{d}^n$ by induction hypothesis, we have that

$$
\begin{aligned}
d^{n+1}(s,a) - \hat{d}^{n+1}(s,a) &\leq \gamma \sum_{s' \in \mathcal{S}} \max_{a' \in \mathcal{A}} \hat{d}^n(s',a') \left( T^a_{ss'} - \hat{T}^a_{ss'} \right) + \\
&\quad - \gamma\epsilon \max_{s',a' \in \mathcal{S} \times \mathcal{A}} \hat{d}^n(s',a') \\
&\leq \gamma \max_{s',a' \in \mathcal{S} \times \mathcal{A}} \hat{d}^n(s',a') \sum_{s' \in \mathcal{S}} \left( T^a_{ss'} - \hat{T}^a_{ss'} \right) + \\
&\quad - \gamma\epsilon \max_{s',a' \in \mathcal{S} \times \mathcal{A}} \hat{d}^n(s',a')
\end{aligned}
$$

Since $\mathbf{Pr}\left( \|T - \hat{T}\|_1 \leq \epsilon \right) \geq 1 - \delta$, we have with probability at least $1 - \delta$,

$$
\begin{aligned}
d^{n+1}(s,a) - \hat{d}^{n+1}(s,a) &\leq \gamma \max_{s',a' \in \mathcal{S} \times \mathcal{A}} \hat{d}^n(s',a')\epsilon - \gamma\epsilon \max_{s',a' \in \mathcal{S} \times \mathcal{A}} \hat{d}^n(s',a') \\
&= 0,
\end{aligned}
$$

which concludes the proof in this case.

Conversely, if $s, a \notin K$, we have

$$
\begin{aligned}
d^{n+1}(s,a) - \hat{d}^{n+1}(s,a) &= D^{M\bar{M}}_{\gamma V^*_{\bar{M}}}(s,a) - \hat{D}^{M\bar{M}}(s,a) + \\
&\quad \gamma \sum_{s' \in \mathcal{S}} T^a_{ss'} \max_{a' \in \mathcal{A}} d^n(s',a') - \gamma \max_{s',a' \in \mathcal{S} \times \mathcal{A}} \hat{d}^n(s',a').
\end{aligned}
$$

Using the same reasoning than in case $s, a \in K$, we have with probability higher than $1 - \delta$

$$
\begin{aligned}
d^{n+1}(s,a) - \hat{d}^{n+1}(s,a) &\leq \gamma \sum_{s' \in \mathcal{S}} T^a_{ss'} \max_{a' \in \mathcal{A}} \hat{d}^n(s',a') - \gamma \max_{s',a' \in \mathcal{S} \times \mathcal{A}} \hat{d}^n(s',a') \\
&\leq \gamma \max_{s',a' \in \mathcal{S} \times \mathcal{A}} \hat{d}^n(s',a') - \gamma \max_{s',a' \in \mathcal{S} \times \mathcal{A}} \hat{d}^n(s',a') \\
&\leq 0,
\end{aligned}
$$

which concludes the proof in the second case. $\qquad\square$

## J   PROOF OF PROPOSITION 6

*Proof.* We follow the proof of the computational complexity of RMax proposed by Strehl et al. (2009). The cost of Lipschitz RMax is constant on most time steps since the action is greedily chosen w.r.t. the upper-bound on the optimal Q-value function which is a lookup table. When updating a new state-action pair (labelling it as a known pair), the algorithm performs $2N$ DP computations to update the Lipschitz bounds plus one DP computation to update the total-bound. The cost of one DP computation is given by (Strehl et al., 2009):

$$
\tilde{\mathcal{O}}\left( SA(S + \log(A)) \frac{1}{1 - \gamma} \log \frac{1}{\epsilon(1 - \gamma)} \right)
$$

The result comes out by remarking that at most $SA$ state-action pairs are updated, each resulting in $(N + 1)$ DP computations. $\qquad\square$

## K   PROOF OF PROPOSITION 7

*Proof.* Consider a fixed state-action pair $s, a \in \mathcal{S} \times \mathcal{A}$. For two sampled tasks $M, \bar{M} \in \hat{\mathcal{M}}^2$, we assume our algorithm to provide an upper-bound on $D^{M\bar{M}}_{\gamma V^*_{\bar{M}}}(s,a)$ with probability at least $1 - \delta$. This assumption is actually guaranteed by Proposition 3 while running Algorithm 1. With probability at least $1 - \delta$,

$$
\hat{D}^{M\bar{M}}(s,a) \geq D^{M\bar{M}}_{\gamma V^*_{\bar{M}}}(s,a), \forall M, \bar{M} \in \hat{\mathcal{M}}^2.
$$

Hence, with probability at least $1 - \delta$,

$$\max_{M,\bar{M}\in\hat{\mathcal{M}}^2} \hat{D}^{M\bar{M}}(s,a) \geq \max_{M,\bar{M}\in\hat{\mathcal{M}}^2} D_{\gamma V_{\bar{M}}^*}^{M\bar{M}}(s,a)$$

i.e. $\hat{D}_{\max}(s,a) \geq D_{\max}(s,a)$.

In turn, the event of underestimating $D_{\max}(s,a)$ occurs only if the two tasks, that we note $M_1^*, M_2^* \in \mathcal{M}^2$, maximizing $M, \bar{M} \mapsto D_{\gamma V_{\bar{M}}^*}^{M\bar{M}}(s,a)$, are not sampled, i.e. do not belong to $\bar{M}$. $M_1^*$ and $M_2^*$ are not necessarily unique, but they could be. Since we aim at deriving a lower bound on the probability of sampling $M_1^*$ and $M_2^*$, we consider the worst case where they are unique. The probability $\tilde{P}$ of sampling one particular task, whose sampling probability is $p$, after $i$ samples, is given by the cumulative distribution function of the geometric distribution and is $p(1-p)^{i-1}$. Consequently, if the sampling probability $p$ of this task is lower bounded by $p_{\min}$, the quantity $p_{\min}(1-p_{\min})^{i-1}$ lower bounds $\tilde{P}$. Let us write $X$ the random variable of the number of samples required for sampling either $M_1^*$ or $M_2^*$ for the first time. By considering that the sampling probability of either sampling $M_1^*$ or $M_2^*$ is lower bounded by $2p_{\min}$, we follow the same reasoning as for $\tilde{P}$ and obtain that :

$$\mathbf{Pr}(X=i) \geq 2p_{\min}(1-2p_{\min})^{i-1}$$

Let us write $Y$ the random variable of the number of samples required for sampling the remaining task for the first time. We have the following result using the geometric distribution for the conditional $\mathbf{Pr}(Y=k|X=i)$:

$$\mathbf{Pr}(Y=k) = \sum_{i=1}^{k-1} \mathbf{Pr}(Y=k, X=i)$$

$$= \sum_{i=1}^{k-1} \mathbf{Pr}(Y=k|X=i)\mathbf{Pr}(X=i)$$

$$\geq 2\sum_{i=1}^{k-1}(1-p_{\min})^{k-i-1}(1-2p_{\min})^{i-1}p_{\min}^2 \qquad (19)$$

$\mathbf{Pr}(Y=k)$ is the probability of first success at step $k$. For $\hat{D}_{\max}(s,a)$ to estimate $D_{\max}(s,a)$ in $m$ steps, we require that this success occurs any time during the first $m$ steps, so we have:

$$\mathbf{Pr}(\hat{D}_{\max}(s,a) \geq D_{\max}(s,a)) = \sum_{k=2}^{m} \mathbf{Pr}(Y=k)$$

Using Equation 19, we can deduce our result when remarking that necessarily $p_{\min} \leq 1/2$:

$$\mathbf{Pr}(\hat{D}_{\max}(s,a) \geq D_{\max}(s,a)) \geq 2p_{\min}^2 \sum_{k=2}^{m}\sum_{i=1}^{k-1}(1-p_{\min})^{k-i-1}(1-2p_{\min})^{i-1}$$

$$\geq 2p_{\min}^2 \sum_{k=0}^{m-2}\sum_{i=0}^{k}(1-p_{\min})^{k-i}(1-2p_{\min})^{i}$$

$$\geq 2p_{\min}^2 \sum_{k=0}^{m-2}(1-p_{\min})^k \sum_{i=0}^{k}\left(\frac{1-2p_{\min}}{1-p_{\min}}\right)^{i}$$

$$\geq 2p_{\min}^2 \sum_{k=0}^{m-2}(1-p_{\min})^k \frac{1}{p}\left(1-p_{\min} - \frac{(1-2p_{\min})^{k+1}}{(1-p_{\min})^k}\right)$$

$$\geq 2p_{\min} \sum_{k=0}^{m-2}\left((1-p_{\min})^{k+1} - (1-2p_{\min})^{k+1}\right)$$

$$\geq 2p_{\min}(1-p_{\min})\frac{1-(1-p_{\min})^{m-1}}{1-(1-p_{\min})}$$

$$- 2p_{\min}(1-2p_{\min})\frac{1-(1-2p_{\min})^{m-1}}{1-(1-2p_{\min})}$$

$$\geq 1 - 2(1-p_{\min})^m + (1-2p_{\min})^m$$

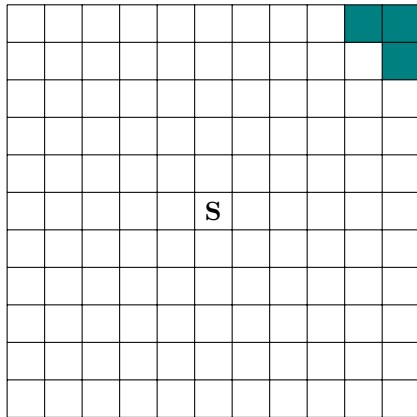

Figure 4: The tight grid-world environment.

□

## L    DISCUSSION ON AN UPPER BOUND ON DISTANCES BETWEEN MDP MODELS

Section 4.3 introduced the idea of exploiting *prior* knowledge on the maximum distance between two MDP models. This idea begs for a more detailed discussion. Consider two MDPs $M$ and $\bar{M}$. By definition of the local model pseudo metric $D_{\gamma V_{\bar{M}}^*}^{M\bar{M}}$ in Equation 1, the maximum possible distance is given by

$$\max_{M, \bar{M} \in \mathcal{M}^2} D_{\gamma V_{\bar{M}}^*}^{M\bar{M}}(s, a) = \frac{1 + \gamma}{1 - \gamma}.$$

But this assumes *any* transition or reward model can define $\bar{M}$. In other words, the maximization is made on the whole set of possible MDPs. To illustrate why this is too naive, consider a game within the Arcade Learning Environment (Bellemare et al., 2013). We, as humans, have a strong bias concerning similarity between environments. If the game changes, we still assume groups of pixels will move together on the screen as the result of game actions. For instance, we generally discard possible new games $\bar{M}$ that "teleport" objects across the screen without physical considerations. We also discard new games that allow transitions from a given screen to another screen full of static. These examples illustrate why the knowledge of $D_{\max}$ is very natural (and also why its precise value may be irrelevant). The same observation can be made for the "tight" experiment of Section 5; the set of possible MDPs is restricted by some implicit assumptions that constrain the maximum distance between tasks. For instance, in these experiments, all transitions move to a neighboring state and never "teleport" the agent to the other side of the gridworld. Without the knowledge of $D_{\max}$, LRMax assumes such environments are possible and therefore transfer values very cautiously. Overall, the experiments of Section 5 confirm this important insight: safe transfer occurs slowly if no a priori is given on the maximum distance between MDPs. On the contrary, the knowledge of $D_{\max}$ allows a faster and more efficient transfer between environments.

## M    THE "TIGHT" ENVIRONMENT USED IN EXPERIMENTS OF SECTION 5

The tight environment is a $11 \times 11$ grid-world illustrated in Figure 4. The initial state of the agent is the central cell displayed with an "S". The actions are moving 1 cell in one of the four cardinal directions. The reward is 0 everywhere, except for executing an action in one of the three teal cells in the upper-right corner. Each time a task is sampled, a slipping probability of executing another action as the one selected is drawn in $[0, 1]$ and the reward received in each one of the teal cells is picked in $[0.8, 1.0]$.

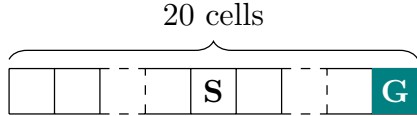

Figure 5: The corridor grid-world environment.

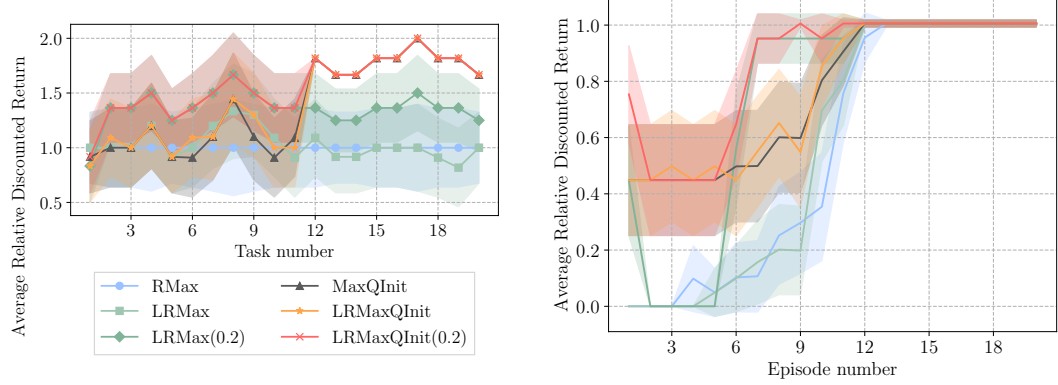

(a) Average discounted return vs. tasks     (b) Average discounted return vs. episodes

Figure 6: Results of the corridor lifelong RL experiment with 95% confidence interval.

## N    ADDITIONAL LIFELONG RL EXPERIMENTS

We ran additional experiments on the corridor grid-world environment represented in Figure 5. The initial state of the agent is the central cell labeled with the letter "S". The actions are {left, right} and the goal is to reach the cell labeled with the letter "G" on the extreme right. A reward $R > 0$ is received when reaching the goal and 0 otherwise. At each new task, a new value of $R$ is sampled in $[0.8, 1]$. The transition function is fixed and deterministic.

The key insight in this experiment is not to lose time exploring the left part of the corridor. We ran 20 episodes of 11 time steps for each one of the 20 sampled tasks. Results are displayed in Figure 6a and 6b, respectively for the average relative discounted return over episodes and over tasks. Similarly as in Section 5, we observe in Figure 6a that LRMax benefits from the transfer method as early as the second task. The MaxQInit algorithm benefits from the transfer from task number 12. Prior knowledge $D_{\max}$ decreases the sample complexity of LRMax as reported earlier and the combination of LRMax with MaxQInit outperforms all other methods by providing a tighter upper-bound on the optimal Q-value function. This decrease of sample complexity is also observed in the episode-wise display of Figure 6b where the convergence happens more quickly on average for LRMax and even more for MaxQInit. This figure allows to see the three learning stages of LRMax reported in Section 5.

We also ran lifelong RL experiments in the maze grid-world of Figure 7. The tasks consists in reaching the goal cell labeled with a "G" while the initial state of the agent is the central cell, labeled with an "S". Two walls configurations are possible, yielding two different tasks with probability $\frac{1}{2}$ of being sampled in the lifelong RL setting. The first task corresponds to the case where orange walls are actually walls and green cells are normal white cells where the agent can go. The second task is the converse, where green walls are walls and orange cells are normal white cells. We run 100 episodes of length 15 time steps and sample a total of 30 different tasks. Results can be found in Figure 8. In this experiment, we observe the increase of performance of LRMax as the value of $D_{\max}$ decreases. The three stages behavior of LRMax reported in Section 5 does not appear in this case. We tested the performance of using the online estimation of the local model distances of Proposition 7 in the algorithm referred by LRMax in Figure 8. Once enough tasks have been sampled, the estimate on the model local distance is used with high confidence on its value and refines the upper-bound computed analytically in Equation 6. Importantly, this instance of LRMax achieved the best result in this particular environment, demonstrating the usefulness of this result. This method being similar

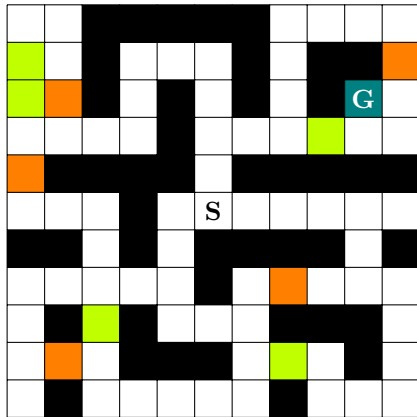

Figure 7: The maze grid-world environment. The walls correspond to the black cells and either the green ones or the orange ones.

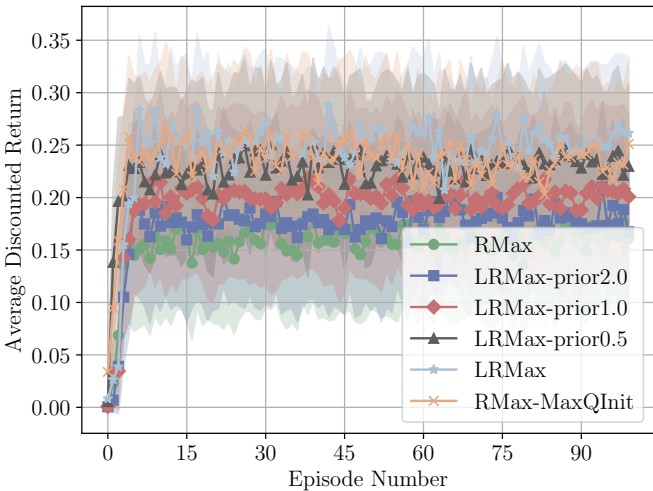

Figure 8: Averaged discounted return over tasks for the maze grid-world lifelong RL experiment.

to the MaxQInit estimation of maximum Q-values, we unsurprisingly observe that both algorithms feature a similar performance in the maze environment.

## O   PRIOR $D_{\max}$ USE EXPERIMENT

Each time an $s, a$ pair is updated, we compute the local distance upper bound $\hat{D}$ (Equation 6) for all $(s, a) \in \mathcal{S} \times \mathcal{A}$. In this computation, one can leverage knowledge of $D_{\max}$ to select $\min\{\hat{D}, D_{\max}\}$. We show that LRMax relies less and less on $D_{\max}$ as knowledge on the current task increases. For this experiment, we used the two grid-worlds environments displayed in Figures 9 and 10.

The rewards collected with any actions performed in the teal cells of both tasks are defined as:

$$R_a^s = \exp\left(-\frac{(s_x - g_x)^2 + (s_y - g_y)^2}{2\sigma^2}\right), \forall s = (s_x, s_y) \in \mathcal{S}, a \in \mathcal{A},$$

where $(s_x, s_y)$ are the coordinates of the current state, $(g_x, g_y)$ the coordinate of the goal cell labelled with a G and $\sigma$ is a span parameter equal to 1 in the first environment and 1.5 in the second environment. The agent starts at the cell labelled with the S letter. Black cells represent unreachable

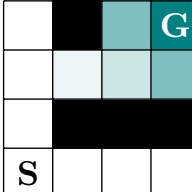

Figure 9: 4 times 4 heat-map grid-world. Slipping probability is 10%.

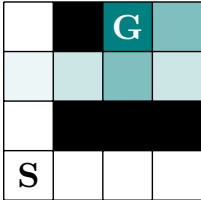

Figure 10: 4 times 4 heat-map grid-world. Slipping probability is 5%.

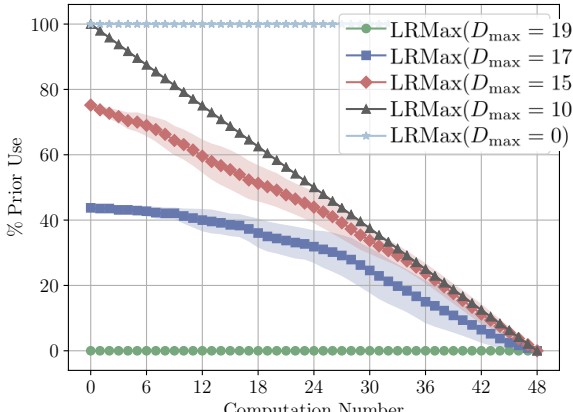

Figure 11: Proportion of times where $D_{\max} \leq \hat{D}^{M\bar{M}}$, i.e. use of the prior, vs computation of the Lipschitz bound. Each curve is displayed with 95% confidence intervals.

cells (walls). We run LRMax twice on the two different maze grid-worlds and record for each model update the proportion of times $D_{\max}$ is smaller than $\hat{D}$ in Figure 11 via the % use of $D_{\max}$.

With maximum value $D_{\max} = 19$, $\hat{D}$ is systematically lesser than $D_{\max}$, resulting in 0% use. Conversely, with minimum value $D_{\max} = 0$, the use expectedly increases to 100%. The in-between value of $D_{\max} = 10$ displays a linear decay of the use. This suggests that, at each update, $\hat{D} \leq D_{\max}$ is only true for one more unique $s, a$ pair, resulting in a constant decay of the use. With fewer prior ($D_{\max} = 15$ or $17$), updating one single $s, a$ pair allows $\hat{D}$ to drop under $D_{\max}$ for more than one pair, resulting in less use of the prior knowledge. The conclusion of this experiment if that $D_{\max}$ is only useful at the beginning of the exploration, while LRMax relies more on its own bound $\hat{D}$ when partial knowledge of the task has been acquired.

# P    DISCUSSION ON RMAX PRECISION PARAMETERS $\epsilon$, $\delta$, $n_{known}$

We used $n_{known} = 10$, $\delta = 0.05$ and $\epsilon = 0.01$. Theoretically, $n_{known}$ should be a lot larger ($\approx 10^5$) in order to reach an accuracy $\epsilon = 0.01$ according to Strehl et al. (2009). However, it is common practice to assume such small values of $n_{known}$ are sufficient to reach an acceptable model accuracy $\epsilon$. Interestingly, empirical validation did not confirm this assumption for any RMax-based algorithm. We keep these values nonetheless for the sake of comparability between algorithms and consistency with the literature. Despite such absence of accuracy guarantees, RMax-based algorithms still perform surprisingly well and are robust to model estimation uncertainties.

## Q    INFORMATIONS ABOUT THE MACHINE LEARNING REPRODUCIBILITY CHECKLIST

For the experiments run in Section 5, the computing infrastructure used was a laptop using a single 64-bit CPU (model: Intel(R) Core(TM) i7-4810MQ CPU @ 2.80GHz). The collected samples sizes and number of evaluation runs for each experiment is summarized in Table 1.

| Task | Number of experiment repetitions | Number of sampled tasks | Number of episodes | Maximum length of episodes | Total number of collected transition samples $(s, a, r, s')$ |
|---|---|---|---|---|---|
| "Tight" task of Figures 1a 1b and 1c | 10 | 15 | 2000 | 10 | 3,000,000 |
| "Tight" task of Figure 1d | 100 | 2 | 2000 | 10 | 4,000,000 |
| Corridor task Section N | 1 | 20 | 20 | 11 | 4400 |
| Maze task Section N | 1 | 30 | 100 | 15 | 45000 |
| Heat-map Section O | 100 | 2 | 100 | 30 | 600,000 |

Table 1: Summary of the number of experiment repetition, number of sampled tasks, number of episodes, maximum length of episodes and upper bounds on the number of collected samples.

The displayed confidence intervals for any curve presented in the paper is the 95% confidence interval (Neyman, 1937) on the displayed mean. No data were excluded neither pre-computed. Hyper-parameters were determined to our appreciation, they may be sub-optimal but we found the results convincing enough to display interesting behaviors.

