# OpenReview forum: "Lipschitz Lifelong Reinforcement Learning"
_ICLR.cc/2020/Conference — Reject_

### Official Review · AnonReviewer3 · 2019-10-22
**Official Blind Review #3**

**Rating:** 6

**Review:**

Updated review: Thanks to the authors for response. I think it clarifies a few points for me except one, which is that I still dont understand how the ATARI games share the same state space? Under what model is this true? And if the paper really wants to use this as an example, why are there no results on even a single ATARI game.

------------------------------------------------------------------------------------------------------------------------------------------------------------
Summary: The paper presents a method for lifelong reinforcement learning problems. By lifelong reinforcement learning the paper means that a set of tasks, specified by MDPs, specified by reward and transition function (R and T) are presented to the agent sequentially and the agent must solve them. The idea of the paper is to assume that the tasks that live in the space <R, T> are Lipschitz continuous and if that is so the experience from the previous task can be used to upper bound the Q values for the current task. The paper suggests using the RMax algorithm that can take as input an upper bound (optimistic initialization) on the Q values of the MDP it is trying to solve and solve the MDP efficiently. For the proposed algorithm called LRMax, the paper presents theoretical results on the sample and computational complexity. The experiments on an 11x11 grid world show that LRMax is more efficient compared to other baselines.

I think this is an interesting paper that takes a principled approach towards a lifelong RL problem. However, I find the paper hard to read and I think that the connection of what paper presents to a real-life problem is missing.

The assumption of Lipschitz continuity of tasks makes little sense to me. Does this mean that the state and action space of the MDPs that specify these tasks have to be the same always? Is that not a very restrictive set of lifelong RL problems? And if it is tried to remove the restriction just by assuming a very big state and action space and how do we verify the Lipschitz continuity and will the method scale to large state, action spaces?

I also found it difficult to understand if the paper is considering an online setting or an offline setting. It seems to me it is considering an online setting, but then proposition 6 and 7 are called sample and computational complexity. In general, in online setting sample complexity can be expressed in terms of T - time step, or the number of rounds. But the paper presents computational complexity in terms of timesteps. Is it just wrong naming or Is the computational complexity the cost of computation at every time step? But then it includes a T term. For quite a lot of time, I thought that this is computational complexity and the T denotes the transition function until I realized, T in this theorem denotes time step.
(Using T for transition function and the time step is confusing).
But then how can computational complexity (that measures the time required) be expressed in terms of time?

The paper claims that it proposes an upper bound on Q values of an MDP that can be computed analytically. I doubt that. First, I think that if this is true then it would have been useful to keep this algorithm in the main body of the paper. I particularly feel that such a bound is a big and important contribution. In any case, it is in the appendix. In the appendix, the paper basically proposes to use dynamic programming to compute the upper bound. So we do assume a model of the transition probabilities of the world that is explicitly maintained and probably learned from the experience of the agent. Does this not restrict the approach of this paper to very small problems for which such tables can be maintained? (This restriction is in addition to the restriction imposed by the assumption of Lipschitz continuity).

Experiments:
The experiments shows that LRMax is more efficient compared to other baselines when it is presented with sequential MDP to solve. This is nice and expected. I think one of the experiments that would have been interesting is what happens when the tasks are not Lipschitz continuous. I would not expect the algorithm the perform well in such cases, but it would be good to know how poorly does the algorithm performs compared to other baselines. So how robust the proposed algorithm is to the assumption it is specifically designed for?

Minor comments:
- the paper keeps using the phrase ‘agent explores greedily wrt to Q function.’ I found this confusing, I think the paper meant agent acts greedily wrt to Q function. The world explore seems to indicate that there is a separate exploration phase.
- The paper claims that distance between two games in arcade learning environment is smaller than the maximum distance between any two MDPs defined on a common state-action space of ALE. Can the paper clarify what this means. Is there a way to verify this claim. What are the common state action space of all games in ALE?
 If it is so, then the LRMax will do great in terms of transferring knowledge in between games on ALE. Why did the paper choose not to show results on ALE then?

**Experience Assessment:**

I have read many papers in this area.

**Review Assessment: Checking Correctness Of Derivations And Theory:**

I assessed the sensibility of the derivations and theory.

**Review Assessment: Checking Correctness Of Experiments:**

I assessed the sensibility of the experiments.

**Review Assessment: Thoroughness In Paper Reading:**

I read the paper at least twice and used my best judgement in assessing the paper.

---

> ### Author Response · Authors · 2019-11-13
> **Answer to Official Blind Review #3**
>
> Dear reviewer,
> Thank you for your thorough reading of the paper and review.
> This is indeed a rather complex topic and we tried to treat it with the necessary rigor, possibly at the cost of clarity.
>
> Based on all three reviews we have uploaded a corrected version of the paper.
>
> We think a few clarifications are needed and would like to try to address your concerns.
>
> Q: "The idea of the paper is to assume that the tasks that live in the space <R, T> are Lipschitz continuous.", "The assumption of Lipschitz continuity of tasks makes little sense to me."
> A: The Lipschitz continuity is not an assumption. We don't assume the MDP to be Lipschitz continuous (as you point out, this makes little sense), we simply state that the Q functions of two "close" MDPs are themselves "close" to each other. This motivates the introduction of the metric on MDP space. The statement above is formalized through the Lipschitz continuity property (proposition 1). This is a direct consequence of the metric's definition, as demonstrated in the appendix.
>
> Q: "Does this mean that the state and action space of the MDPs that specify these tasks have to be the same always? Is that not a very restrictive set of lifelong RL problems?"
> A: Yes, the state and action spaces are the same, as in most recent work on the matter [1,2,4]. We believe it is a very general case: Atari games share the same state-action space. So do the MDPs defined for different robotic tasks performed by the same robot [5].
>
> Q: "I also found it difficult to understand if the paper is considering an online setting or an offline setting."
> A: We consider an online Lifelong RL setting, as in [1]. An offline setting would boil down to multi-task RL but without the sequence of MDPs component which makes the difference with Lifelong RL. We took great care to avoid any confusion on this topic by explictly mentioning it at several places in the updated version.
>
> Q: "In general, in online setting sample complexity can be expressed in terms of T - time step, or the number of rounds. But the paper presents computational complexity in terms of timesteps."
> A: Following [3], we express the sample complexity in Proposition 6 as an upper-bound on the total number of samples, and thus interaction time steps, before reaching an epsilon-optimal behavior. Thank you for pointing out the confusion of notation for this number of interaction steps: we wrote it $\tau$ instead of $T$. In turn, the computational complexity (Proposition 7) is an upper bound on the total number of operations needed by a computer to run LRMAX for $\tau$ time steps. We rephrased the two propositions accordingly to avoid any confusion.
>
> Q: "The paper claims that it proposes an upper bound on Q values of an MDP that can be computed analytically."
> A: We do not claim that the upper bound $\hat{U}$ can be computed analytically. The only thing for which we provide and analytical expression is $\hat{D}$ of Equation 7 (proof of the derivation in Section H of the Appendix). This expression, in turn, is used in the Dynamic Programming resolution of Equation 8, which finally leads directly to the upper-bound $\hat{U}$.
>
> Q: "So we do assume a model of the transition probabilities of the world that is explicitly maintained and probably learned from the experience of the agent."
> A: Rmax is a tabular model-based algorithm (and all recent papers on the matter are framed in this same tabular contact). We believe the primary contribution of this paper is introducing this non-negative transfer method in a well understood case. Although scaling up to large instances is indeed a valuable and much desirable extension, it will require several additions such as moving to continuous state spaces (the extension of Rmax to such a case is non-trivial), discarding the model (by using delayed Q-learning for instance) and introducing function approximation. All these extensions are important future work and it seems detrimental to mix all these contributions in the same conference paper.
>
> References:
> [1] David Abel, Yuu Jinnai, Sophie Yue Guo, George Konidaris, and Michael L. Littman. Policy and Value Transfer in Lifelong Reinforcement Learning. In International Conference on Machine Learning, 2018.
> [2] Emma Brunskill and Lihong Li. Pac-inspired option discovery in lifelong reinforcement learning. In International Conference on Machine Learning, 2014.
> [3] Alexander L. Strehl, Lihong Li, and Michael L. Littman. Reinforcement learning in finite MDPs: PAC analysis. Journal of Machine Learning Research, 10(Nov):2413–2444, 2009.
> [4] Aaron Wilson, Alan Fern, Soumya Ray, and Prasad Tadepalli. Multi-task reinforcement learning: a hierarchical Bayesian approach. In International Conference on Machine Learning, 2007.
> [5] Tianhe Yu, Deirdre Quillen, Zhanpeng He, Ryan Julian, Karol Hausman, Chelsea Finn and Sergey Levine. Meta-World: A Benchmark and Evaluation for Multi-Task and Meta Reinforcement Learning. CoRL, 2019.

---

### Official Review · AnonReviewer1 · 2019-11-07
**Official Blind Review #1**

**Rating:** 3

**Review:**

This paper studies the problem of reusing prior experience in Reinforcement learning. The main contribution of this work is to introduce a novel metric between MDPs. They use the value functions Lipschitz continuity in the task space w.r.t the metric and show the connection. The theoretical results of this paper show the value transfer and they can derive results for the same. They demonstrate the benefit of their method in experiments.

Pros of this paper:
- The idea of using the Lipschitz continuity of the value seems interesting
- The paper seems to solve an important problem

Cons of this work
- I think this paper is not yet ready for publication. It is not clearly written and the intuitions does not come up clearly through the writing
- The intuition and the utility of this work is hidden behind the math and the theory
- What are the key insights which a practitioner can derive from this?
- The experiments do not seem realistic and rather contrived
- Because of the mathematical density, the motivation of Lipschitz continuity  is not very clear
- Lot of important details are not clear from the write-up. Is this the offline or the online setting (pointed out by the other reviewer as well).

Overall, I think this paper is an important contribution. However, in its current form it does not seem accessible and lacks a lot of the motivation,

**Experience Assessment:**

I have read many papers in this area.

**Review Assessment: Checking Correctness Of Derivations And Theory:**

I assessed the sensibility of the derivations and theory.

**Review Assessment: Checking Correctness Of Experiments:**

I assessed the sensibility of the experiments.

**Review Assessment: Thoroughness In Paper Reading:**

I read the paper at least twice and used my best judgement in assessing the paper.

---

> ### Author Response · Authors · 2019-11-13
> **Answer to Official Blind Review #1**
>
> Dear reviewer,
> thank you very much for your comments. This is indeed a rather complex topic and we tried to treat it with the necessary rigor, possibly at the cost of clarity. We very gladly welcome all comments that improve the clarity and didacticism.
>
> Based on all three reviews we have uploaded a corrected version of the paper.
>
> We tried to correct the presentation to make it clearer. Given the paper size constraints, we did our best to better introduce the motivations at every step and the reasoning process. This led to adding some explanatory sentences, better motivating the introduction of definitions and propositions, or sometimes refactoring a paragraph (at the beginning of Section 4 for instance). We also tried to highlight the high-level conclusions.
>
> In hindsight, we believe this contribution provides a sound basis to non-negative value transfer via MDP similarity, a development that was lacking in the literature. Key insights for the practitioner lie both in the theoretical analysis and in the practical derivation of a transfer scheme that achieves non-negative transfer with PAC guarantees. The previous two sentences were added to the conclusion (on top of several rephrasings in the paper and in the conclusion in particular).
> Another way of phrasing these insights is in understanding finely what is the actual link between MDP similarity and provable value transfer. One of the key findings that we tried to illustrate both theoretically and empirically is the strong bias we, as humans, have towards similarity between environments. Due to lack of space in the paper, a short discussion on the matter has been included in the Appendix, Section L.
>
> How we addressed your comments in detail:
> Q: "The intuition and the utility of this work is hidden behind the math and the theory"
> A: Thanks for pointing this out. We tried to remain as rigorous as possible and we realize that was detrimental to the clarity of the paper. We hope to have corrected this in the updated version (small edits all along the paper that aim at giving better motivation and a clearer train of thoughts by highlighting key ideas).
>
> Q: "What are the key insights which a practitioner can derive from this?"
> A: We felt this should be emphasized in the conclusion. So, on top of the aforementioned edits, we added a sentence in the conclusion to state these insights clearly.
>
> Q: "The experiments do not seem realistic and rather contrived"
> A: Our goal was to provide a sound basis to value transfer, both theoretically and practically. But scaling up to realistic, large scale instances, is a whole other contribution since it will require several additions such as moving to continuous state spaces (the extension of Rmax to such a case is non-trivial) and introducing function approximations. All these extensions are important future work and it seems detrimental to mix all these contributions in the same conference paper. Nevertheless, as pointed also by review 4, additional experiments do strenghten the paper. We re-included some experiments that we had taken out earlier (for clarity reasons), they can be found in the Appendix, Section N (and are refered to in the main text).
>
> Q: "Because of the mathematical density, the motivation of Lipschitz continuity is not very clear"
> A: We hope to have clarified how Lipschitz continuity is just a proxy notion for "two similar MDPs have similar Q^* functions".
>
> Q: "Lot of important details are not clear from the write-up. Is this the offline or the online setting (pointed out by the other reviewer as well)."
> A: Thanks again for pointing this out. This is really the online setting, just as vanilla RMax. We took great care to avoid any confusion on this topic by explictly mentioning it at several places in the updated version.

---

### Official Review · AnonReviewer4 · 2019-11-08
**Official Blind Review #4**

**Rating:** 6

**Review:**

Edit: in light of the discussion below, I have decided to raise my score to a 6.

* Summary
This paper introduces a pseudo-metric based on a notion of Lipschitz continuity for action-value functions between different MDPs.
The algorithmic improvement is the use of the pseudo metric with RMax in lifelong RL.
In particular, the pseudo-metric is defined as the minimum between two dissimilarity measures, each of which can be solved with dynamic programming.
The pseudo metric is then used in conjunction with the optimal state-action value function for a previous MDP to form an upper bound heuristic.
This upper bound is used in RMax to improve exploration since it can be tighter than the naive RMax bound.
Empirically, the theoretical results are supported by thorough investigation on a variant of the grid-world environment.


* Decision
The main contribution of this paper is hard to discern, but the ideas presented are interesting.
Despite the theoretical nature of the paper, I find that the experimentation is overall lacking.
In the paper's current state, I would recommend a weak rejection.
However, I am willing to increase this score if the presentation is improved and if the experiments are expanded.


* Reasons
While the paper motivates and explains the problem well, the proposed method is much less clear.
For example, Section 2 leading up to proposition 3 seems like it should motivate Section 3.
Instead, I think that Section 2 does not communicate enough how the pseudo-metrics are planned to be used.
Perhaps proposition 3 can be in the appendix and RMax should be discussed in section 2.
Otherwise, the pseudo-metrics in Section 2 (and 3) should be motivated some other way.

For experiments, the paper provides a thorough investigation of one interesting environment.
I think a similarly thorough investigation of another environment would greatly strengthen the paper.
Multiple experiments that are less thorough would also benefit the paper, but less so.
Lastly, it would be interesting to compare to other non-PAC-MDP lifelong learning RL algorithms.


* Details

Minor concerns:

I have trouble following the statement and proof of your probability bounds (for example, proposition 4).
It would be good to make more explicit what is a random variables.
For example, you define R_s^a to be the expected reward, but I assume that $(s,a)$ are random variables in the probability bound.
In addition, you occasionally refer to the propositions and equations as properties.

Page 2: "The extension to continuous spaces is straightforward but beyond the scope"
This does not seem relevant since you only look at the tabular problem.
While the definition might hold for continuous state spaces, the extension of LRmax to continuous spaces does not seem straightforward.

Minor typos:
Page 4: "this information when the task changes allows to compute the upper bound"


**Experience Assessment:**

I have read many papers in this area.

**Review Assessment: Checking Correctness Of Derivations And Theory:**

I assessed the sensibility of the derivations and theory.

**Review Assessment: Checking Correctness Of Experiments:**

I carefully checked the experiments.

**Review Assessment: Thoroughness In Paper Reading:**

I read the paper at least twice and used my best judgement in assessing the paper.

---

> ### Author Response · Authors · 2019-11-13
> **Answer to Official Blind Review #4**
>
> Dear reviewer,
> thank you very much for your comments. This is indeed a rather complex topic and we tried to treat it with the necessary rigor, possibly at the cost of clarity. We very gladly welcome all comments that improve the clarity and didacticism.
>
> Based on all three reviews we have uploaded a corrected version of the paper.
>
> We tried to correct the presentation to make it clearer. Given the paper size constraints, we did our best to better introduce the motivations at every step and the reasoning process. This led to adding some explanatory sentences, better motivating the introduction of definitions and propositions, or sometimes refactoring a paragraph (at the beginning of Section 4 for instance). We also tried to highlight the high-level conclusions.
>
> In a previous version of the paper, we also reported on extra experiments to better illustrate how LRMax behaves in various environments. We previously took them out of the paper to unclutter the discussion, gain space, and focus on specific points in the discussion. We agree these should have been mentioned: these experiments (corridor, larger maze) are back in the Appendix (section N).
>
> How we addressed your comments in detail:
> Q: "While the paper motivates and explains the problem well, the proposed method is much less clear."
> A: We have updated Section 4 so that the reasoning leading to Lipschitz RMax is better motivated. Particularly, we emphasized high level ideas.
>
> Q: "For example, Section 2 leading up to proposition 3 seems like it should motivate Section 3."
> A: We are unsure of what you are refering to, since proposition 3 is in Section 4. We did out best to better motivate the reasoning in general and hope our corrections addressed any lack of clarity you noticed.
>
> Q: "Instead, I think that Section 2 does not communicate enough how the pseudo-metrics are planned to be used."
> A: We clarified this specific point in the first paragraph of Section 3.
>
> Q: "Perhaps proposition 3 can be in the appendix and RMax should be discussed in section 2."
> A: We would rather keep Proposition 3 in the paper since it serves the discussion later on (and is refered to in Algorithm 1). We hope this helps keeping the paper self-contained (and the Appendix is already 15 pages long).
>
> Q: "Otherwise, the pseudo-metrics in Section 2 (and 3) should be motivated some other way."
> A: We hope the improved introductory paragraph of section 3 clarifies this.
>
> Q: "For experiments, the paper provides a thorough investigation of one interesting environment. I think a similarly thorough investigation of another environment would greatly strengthen the paper. Multiple experiments that are less thorough would also benefit the paper, but less so."
> A: Since the paper was already dense, we restored previous results and highlighted the related key insights in the Appendix, Section N (this resulted in shifting subsequent sections of the Appendix).
>
> Q: "Lastly, it would be interesting to compare to other non-PAC-MDP lifelong learning RL algorithms."
> A: We agree on this point. As a matter of fact, we conclude on the comparison with other PAC-MDP algorithms but comparisons with non-PAC-MDP methods is definitely relevant. We believe however this is a separate contribution since it implicitly asks the question of PAC bounds for non-PAC-MDP algorithms. It is also coupled with the question of scaling-up to large instances which is an important (but separate) next step (we discuss this in our answer to reviews 1 and).

---

> > ### Comment · AnonReviewer4 · 2019-11-15
> > **Comments on Proposition 3 should have been Proposition 2**
> >
> > Dear authors,
> >
> > The updated version has addressed many of my concerns.
> >
> > I apologize for the confusing comments regarding proposition 3, which should have been about proposition 2 (global lipschitz continuity). I do not think that proposition 2 helps motivate subsequent sections, nor does it help in understanding previous sections. I understand that it is noted for its theoretical interest, but I believe that might be better suited for the appendix.

---

> > > ### Author Response · Authors · 2019-11-15
> > > **Proposition 2 moved to the appendix**
> > >
> > > Dear reviewer,
> > > thanks a lot (again). Keeping this result in the main body of paper was something we had mixed feelings about in the first place. Moving it to the appendix now allows a much smoother and logical transition from Section 3 to Section 4.

---

### Decision · Program_Chairs · 2019-12-19

**Decision:**

Reject

**Comment:**

While there was some support for this paper, there was not enough support to accept it for publication at ICLR.

The following concern is characteristic of the concerns raised by the reviewers: "The "main contribution of this paper is hard to discern, but the ideas presented are interesting." Other reviewers said it was "hard to read" and not ready for publication.